# Place recognition using batlike sonar

**Dieter Vanderelst[1,2]\*, Jan Steckel[2,3], Andre Boen[2], Herbert Peremans[2], Marc W Holderied[1]**

[1]School of Biological Sciences, University of Bristol, Bristol, United Kingdom; [2]Active Perception Lab, University of Antwerp, Antwerp, Belgium; [3]Constrained Systems Lab, Faculty of Applied Engineering, University of Antwerp, Antwerp, Belgium

**Abstract** Echolocating bats have excellent spatial memory and are able to navigate to salient locations using bio-sonar. Navigating and route-following require animals to recognize places. Currently, it is mostly unknown how bats recognize places using echolocation. In this paper, we propose template based place recognition might underlie sonar-based navigation in bats. Under this hypothesis, bats recognize places by remembering their echo signature - rather than their 3D layout. Using a large body of ensonification data collected in three different habitats, we test the viability of this hypothesis assessing two critical properties of the proposed echo signatures: (1) they can be uniquely classified and (2) they vary continuously across space. Based on the results presented, we conclude that the proposed echo signatures satisfy both criteria. We discuss how these two properties of the echo signatures can support navigation and building a cognitive map.

## Introduction

Echolocating bats have excellent spatial memory (*Barchi et al., 2013*; *Holland, 2007*; *Geva-Sagiv et al., 2015*) and are able to navigate to salient locations like roosts, foraging grounds and drinking places (*Schnitzler et al., 2003*). Experimental results obtained using both *Phyllostomus hastatus* and *Rousettus aegyptiacus* suggest that long distance navigation and migration seem to be supported mainly by visual cues (*Tsoar et al., 2011*; *Holland, 2007*). However, when displaced by less than about 15 km, both *Myotis* spp. and *Phyllostomus hastatus* deprived of sight have been found to successfully return to their roost (*Stones and Branick, 1969*; *Williams et al., 1966*). This shows that navigation in bats can be supported by echolocation as well as vision. Navigating and route-following require animals to recognize places (*Franz and Mallot, 2000*). Currently, it is largely unknown how bats recognize places using echolocation (*Schnitzler et al., 2003*) and different mechanisms are possible (*Geva-Sagiv et al., 2015*).

### Model based place recognition

One possible mechanism for sonar based recognition of places is to localize and classify the individual objects in the scene observed from that place is composed of (*Lewicki et al., 2014*). Under this assumption, bats would reconstruct a (presumably, 3D) model of the layout of a given place and match this with a set of previously stored representations, e.g. (*Barchi et al., 2013*; *Moss and Surlykke, 2001*; *Schnitzler et al., 2003*). In other words, this mechanism implies that bats reconstruct a mental image of the environment from the echoes and use this to recognize a previously visited place.

Under favourable conditions, extracting a description of a set of localized objects from the echoes is possible as the position, shape, size and texture of objects are all encoded in the binaural spectra of the echoes, e.g. (*Reijniers et al., 2010*; *Schmidt, 1988*; *Von Helversen and Von Helversen, 2003*; *Peremans et al., 2012*; *Wotton and Simmons, 2000*; *Simon et al., 2011*; *Genzel and Wiegrebe, 2013*). However, a number of limitations of bio-sonar render it uncertain

\*For correspondence: dieter.
vanderelst@uantwerpen.be

**Competing interests:** The authors declare that no competing interests exist.

**eLife digest** Bats produce loud calls and listen to the returning echoes to find their way around. This process, known as echolocation, is sometimes described as 'seeing with sound'. The way bats perceive the world through echolocation, however, is fundamentally different from how we experience it through vision. Echolocation provides much less information about the world than vision does, but despite this, bats are agile navigators and hunters.

It is not clear how bats navigate so well without much information. In particular, researchers would like to know how echolocating bats recognize the places that they regularly visit while foraging and navigating. When we visually recognize places, we identify and localize the various objects making up the scene. But echolocation is unlikely to provide enough information to allow bats to identify and localize the objects in a particular place.

To investigate how bats recognize places, Vanderelst et al. built an artificial bat: a device that contained ultrasonic microphones to act like the bat's ears and an ultrasonic speaker to act like the bat's mouth. The artificial bat device was then used to collect echoes from different locations in real bat habitats.

Processing the echoes using machine-learning techniques showed that the echoes that returned from each location were different enough for a computer to recognize the location. By using a simplified version of the echoes, Vanderelst et al. also showed that the locations could be recognized even if there was not enough information to identify specific objects or vegetation at the site. This suggests that bats do not simply use echolocation to recreate the three-dimensional layout of a location, as some researchers have proposed.

While much remains to be learned about how bats use echolocation for navigation, future work that teases out bat navigation strategies might help us to build robots that can navigate using similar tactics.

whether bats are able to apply such a place recognition strategy. Low signal-to-noise ratio obscures echo spectra (*Reijniers et al., 2010*). Moreover, ambiguity of cues is introduced by concurrently encoding in the spectral cues both location and other properties of objects. For example, a spectral notch could be introduced by either the head related transfer function (encoding the object's location) or the transfer function of the reflector (encoding the object's shape). This complicates extracting both object properties and object positions. Reconstructing the 3D position of objects is further complicated by the fact that, compared to vision, sonar has a limited field of view (*Surlykke et al., 2009*), a low update rate (*Kleeman and Kuc, 2008*; *Peremans et al., 2012*) and a limited range (*Stilz and Schnitzler, 2012*). While most of these limitations to reconstructing the layout of objects could potentially be addressed by integrating information across calls, this would entail both time and complexity penalties.

In addition, the capacity for reconstructing the 3D layout of objects is limited by the finite temporal resolution of the bat's auditory system. The temporal integration in the auditory system severely limits the spatial resolution of biosonar and introduces interference between echoes. In a psychophysical experiment, *Wiegrebe and Schmidt (1996)* observed a temporal integration constant of about 200 $\mu$s in *Megaderma lyra*. Simmons and colleagues (*Simmons et al., 1989*) derived an integration constant of about 200–400 $\mu$s for the bat *Eptesicus fuscus*. Echoes from objects separated in time by less than the response time of the auditory filters will be integrated (*Simmons et al., 1989*), thereby limiting bats' ability to resolve the objects' locations and properties.

The limitations imposed by temporal integration can be appreciated by considering the volume of space across which echoes are integrated by the sonar system. This volume can be appropriately described as a section of a spherical shell. The thickness of the shell is determined by the temporal integration of the hearing apparatus. The opening angle of the section is determined by the directionality of the sonar system. Hence, the integrated volume increases rapidly for larger distances from the bat (See *Figure 1* and equation therein). For example, at a distance of 7.8 meters, an echolocation system with a beamwidth of 45 degrees is estimated to integrate the echoes originating from a volume of over 1 m³. At the same distance, a system with a functional beamwidth of 60

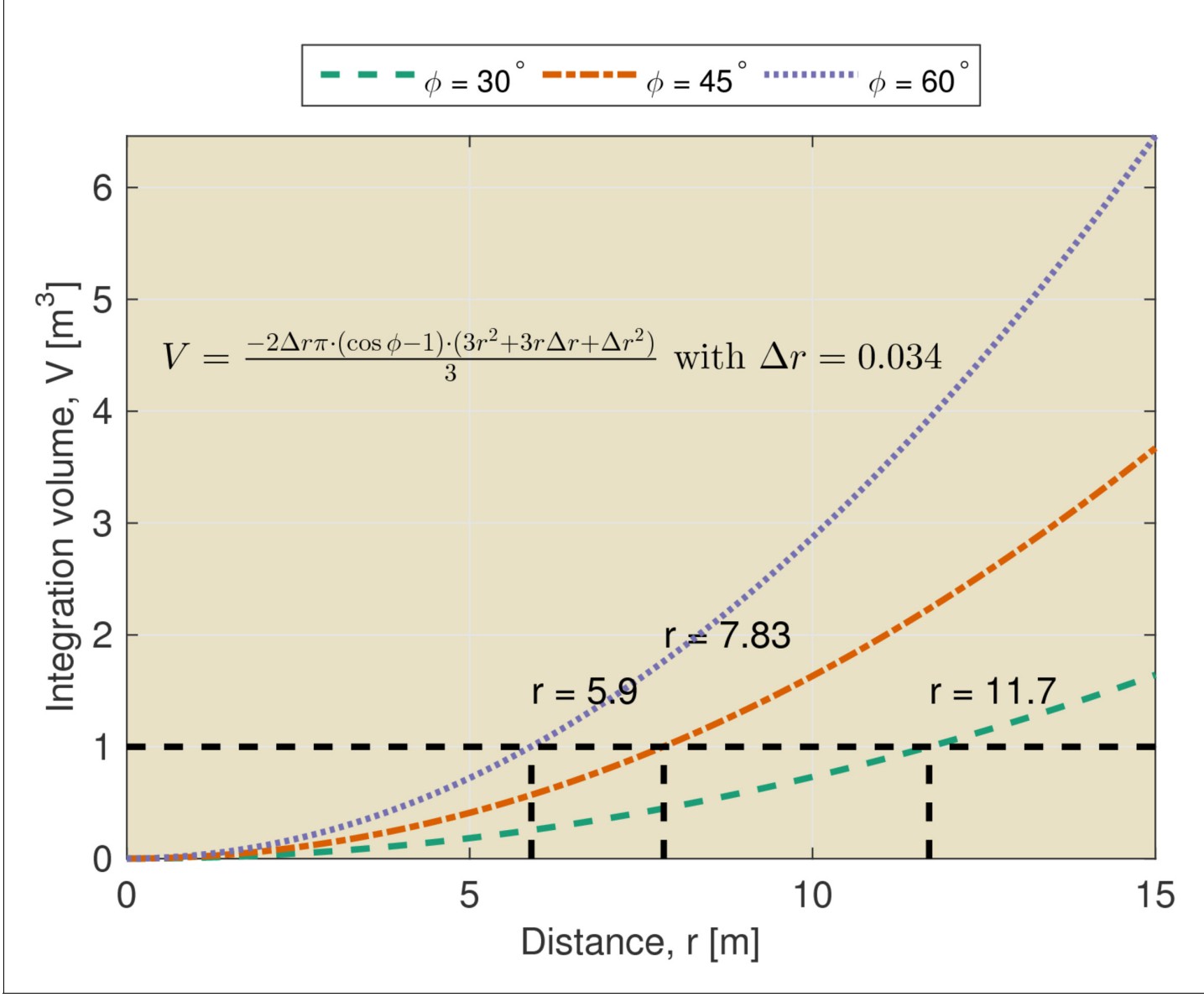

**Figure 1.** Integration volume as a function of distance. The integration volume calculated for different beamwidths $\phi$ as a function of distance from the bat. The integration volume is approximated as a section of a spherical shell. The opening angle of the section is given by the beam width $\phi$. The thickness of the shell $\Delta r$ is given by the temporal integration of the sonar system. The temporal integration of the sonar system has been assumed to be 200 $\mu$s yielding an integration distance (i.e. thickness of the shell) of about 0.034 m.

degrees integrates echoes from a volume of about 2 m$^3$. Echoes originating from reflectors within this volume will be integrated and it is not clear that they will be perceived as individual echoes (*Geberl, 2013*). Indeed, in general, this volume will contain multiple reflectors. This is especially true when the scene contains complex objects such as leafy trees or bushes (See *Yovel et al., 2008* for examples of echo trains returning from vegetation).

Finally, it should be noted that the integration constant used to compute the volume of the sphere represents a very conservative upper boundary for the temporal resolution of the auditory system. Indeed, non-simultaneous masking effects have been found to extend for substantially longer intervals than the integration time (both in bats and humans [*Moore, 2012*]). For example, *Geberl (2013)* found non-simultaneous masking in *Phyllostomus discolor* for temporal separation between echoes up to 6 ms.

Some theoretical mechanisms, e.g. (*Fontaine and Peremans, 2009*; *Matsuo et al., 2004*; *Saillant et al., 1993*), have been proposed that would allow bats to extract environment impulse responses from signals composed of multiple closely spaced echo signals. However, none of them have been unequivocally proven to be exploited by bats. Moreover, so far, these mechanisms have only been tested using simple artificially generated echo trains. To the best of our knowledge, no algorithm has proven to be able to reconstruct the impulse response of a realistically complex reflector such as vegetation.

*Grunwald et al. (2004)* reported that bats can discriminate between echoes with different impulse response statistics. This shows that the auditory periphery of the bat retains sufficient information to allow responding to the stochastic properties of the underlying impulse responses. However, this does not necessarily imply that bats can reconstruct the impulse response from the echoes, which is required to be able to reconstruct the 3D layout of a scene from the echoes.

In summary, inherent limitations of biosonar, including (1) the dependence on ambiguous spectral cues for both object localization and classification and (2) the finite temporal resolution of the auditory system, render it unlikely that an explicit 3D representation of a complex scene is available to a cruising bat – and that such a representation is used to recognize places.

As the capacity for recognizing places is nevertheless a necessary requirement for any navigation strategy, we propose an alternative mechanism, taking into account these limitations, to explain how echolocating bats might navigate.

## Template based place recognition

Instead of attempting to reconstruct the position, shape and identity of objects from the cochlear output, we propose the cochlear output to be used directly, extending the template based approach described by *Wiegrebe (2008)*. Under this hypothesis, bats are assumed to match the output of the cochlea to a set of stored templates, i.e., classifying the echo signature as one of a set of memorized echo signatures each one recorded at a previously visited place. This approach obviates the need for complex reconstruction algorithms extracting 3D spatial information from the echo signals at the two ears. As it uses the sensory input directly, our approach to place recognition is analogous to the view-based place recognition that is thought to underlie visual navigation in insects (*Zeil et al., 2003*; *Franz and Mallot, 2000*).

In addition to circumventing the computationally hard problem of deriving a 3D spatial representation from complex echo signals, echo templates have been shown to be very discriminative. *Kuc (1997)* showed a bio-mimetic sonar device to be capable of detecting which side of a coin was up using very simple echo templates. Their computational simplicity and discriminative power have led sonar based templates being used to recognize places in robotic navigation algorithms before, e.g. (*Mataric and Brooks, 1990*; *Steckel and Peremans, 2013*; *Kuipers, 2000*). However, to the best of our knowledge, no study has looked at the properties of sonar templates accessible to bats operating in complex habitats. Indeed, robotic studies using sonar templates have all been carried out in artificial man-made environments. Moreover, these studies have used emitters, receivers and processing methods that are not necessarily biologically plausible. For example, robots typically (*Kleeman and Kuc, 2008*) use a ring of sonar ranging devices (*Mataric and Brooks, 1990*; *Kuipers, 2000*), each with a limited field of view (*Steckel and Peremans, 2012*), that extract the delay of the first echo.

In this paper, using ensonification data, we derive templates from many echo trains from natural habitats. In contrast to most robotic studies, we collect and process the echo data in a biologically plausible way. Next, we assess the viability of our hypothesis by evaluating whether the derived templates are sufficient to support place recognition. Indeed, the viability of a template-based approach to place recognition depends on the following two properties of the templates:

1. Templates must allow for unique classification for places to be recognizable. In other words, templates must encode specific locations and orientations of the bat in space. If templates can not be uniquely classified, they can not be used to recognize previously visited places.
2. Templates should vary smoothly as a function of the bat's location and orientation. The dissimilarity between templates should increase monotonically over a relevant (non-trivially small) distance and angle. This allows the bat to recognize (that it is near) a place even if it is not exactly in the same location or orientation it was before.

In this paper, we test whether these conditions are satisfied by collecting many cochlear templates as they could be perceived by bats. In the discussion, we argue that, by satisfying these two criteria, the templates can support navigation. In addition, we discuss how the templates could support the acquisition of a cognitive map.

## Materials and methods

All data and computer scripts are available from Zenodo (*Vanderelst et al., 2016*).

### Ensonification

A custom ensonification device consisting of 31 Knowles FG series microphones (*Steckel and Peremans, 2012*) and a Senscomp (http://www.senscomp.com/ultrasonic-sensors/series-7000-sensors.php) Series 7000 ultrasonic speaker was used (see *Figure 2*). The speaker produces about 106 $dB_{spl}$ at 50 kHz and 1 meter distance. The device was mounted on a pan tilt system (PTU-E46, FLIR, Goleta, CA) allowing us to rotate the device from −150 to 150 degrees in azimuth and from −25 to 40 degrees in elevation. The pan tilt system moved the ensonification device through the full extent of its mechanical range in 31 steps of 10 degrees in azimuth and 7 steps of 10.8 degrees in elevation The positional error of the pan tilt system is less than 0.1 degrees.

At each azimuth and elevation direction, three measurements were gathered. The data capture and storage was handled by a single-board computer integrated into the ensonification device. The single board computer controlling the data collection also controlled the pan tilt system through a serial interface cable. The single-board computer was wirelessly connected to a laptop enabling us to start and monitor the data collection. A hyperbolic frequency modulated pulse sweeping from 100 to 40 kHz in 1 ms was emitted. This frequency range was limited by the frequency response of the emitter and the electronics. The recording of the echoes was started at the onset of the emission and ends 34 ms later. The duration of the recording time window was limited by the on-board memory of the data acquisition system. The sampling rate was 219 kSamples/s.

Data was collected at three different sites (see *Figure 3*). The structure of the data is depicted in *Figure 3*. First, twelve positions in St. Andrews Park (Bristol, UK, 51:27:15.772N, 2:36:52.996W) were selected for ensonification. To ensure the data represented different densities of clutter bats might operate in, four open, four semi-cluttered and four cluttered locations were selected. The data collected at St. Andrews Park consisted of 7812 (12 positions × 217 directions × 3 repeats) echo trains at each of the 31 microphones. Second, data was collected in a park in Midreshet Ben Gurion, Israel (30:50:53.318N, 34:46:54.174E). At this site, bats use an artificial corridor lined with boulders as part of their commuting route. In this corridor, the ensonification device was placed at 50 positions along a straight line spaced 20 cm apart. The line approximately ran along the centre of the corridor. At each of the 50 positions, three measurements for each of the 217 directions were collected yielding 32,550 echo trains (50 positions × 217 directions × 3 repeats) at each of the 31 microphones. Finally, data was collected at Royal Fort Gardens (Bristol, UK, 51:27:26.417N, 2:36:4.619W). At this third site, 40 positions spaced 25 cm apart along a 10 meter line were sampled yielding 26,040 echo trains (40 positions × 217 directions × 3 repeats) at each of the 31 microphones. In total, 66,402 echo trains for each of the 31 microphones were used in this paper, resulting in a data set consisting of a total of 2,058,462 echo trains.

At both St. Andrews park and the Israel site data was collected at an approximate height of about 3 meters. In the case of St. Andrews park, overhanging vegetation sometimes restricted the device to be raised to this level. If so, the device was raised as high as possible. In the Royal Fort Park, all data was collected at a height of about 2 meters.

### Template construction

Templates were derived from the data collected at each of the three sites. The data for each of the 12, 50 and 40 positions in each data set were processed in the same manner. The method used in constructing the templates is summarized in algorithm 1.

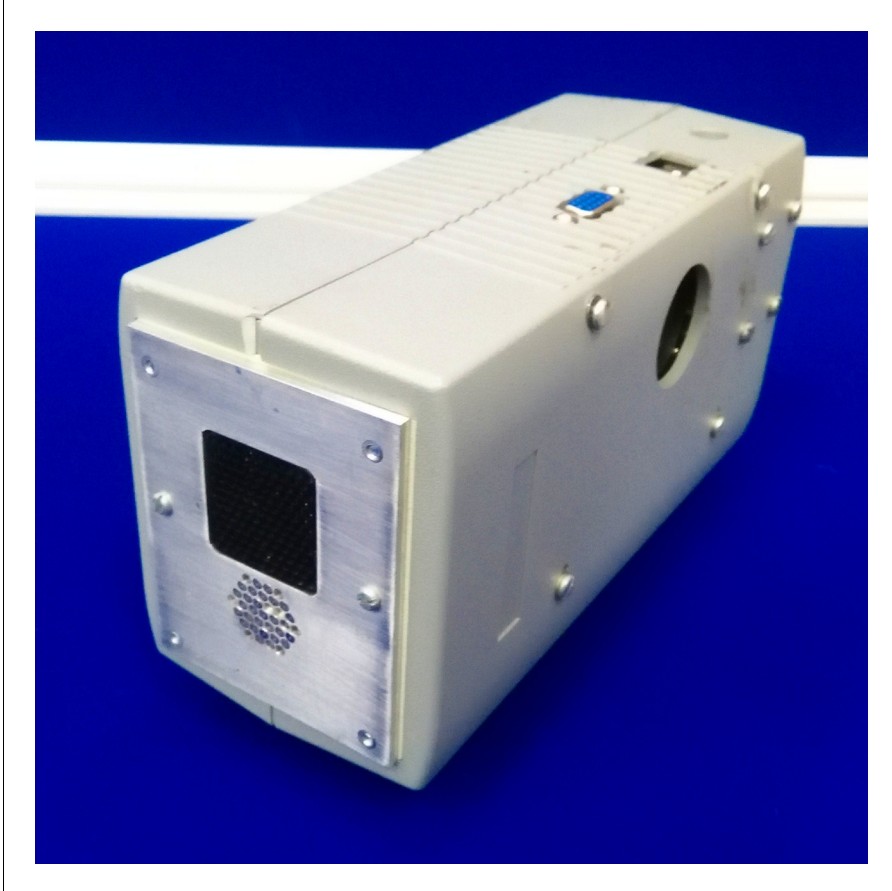

**Figure 2.** Close up of the ensonification device. At the front of the device the array of 31 microphones (Knowles FG series) can be seen. The Senscomp Series 7000 ultrasonic speaker is located above this array.

## Algorithm 1: The template construction process

For each position, the data consisted of 20,181 echo trains collected from 217 azimuth and elevation directions. At each azimuth and elevation direction, 3 measurements were taken using 31 microphones (217 directions × 31 microphones × 3 repeats = 20,181). Step 1: Each of the 20,181 echo trains was filtered using a model of the bat's auditory periphery. This model returns a cochleogram. Step 2: The first 5.8 ms were set to zero to avoid the pick up of the emitted signal to be processed. Step 3: The cochleograms returned by the model of the auditory periphery were averaged across frequency and the 31 microphones. This resulted in 3 (corresponding to the 3 r) templates for each of the 217 azimuths & elevation directions. At this point, the data consisted of 651 templates, corresponding to 3 repeats for each of the 217 directions. Step 4: To obtain a realistic directionality, the templates were averaged across 9 neighboring azimuth and elevation directions (See *Figure 4* for a depiction of the resulting virtual directionality). Step 5: The templates were sampled at an interval of 350 $\mu$s. Step 6: Finally, templates were averaged across the 3 repeats to obtain a single template for each of the 217 directions.

1. **Input: Data for a single position: 20,181 echo trains**
2. Step 1: Cochlear model and dechirping
3. Step 2: Set first 5.8 ms to zero
4. Step 3: Average across frequencies and microphones
5. **Intermediate result: 651 templates** (217 templates × 3 repeats = 651)
6. Step 4: Average across 3 × 3 neighbouring directions
7. Step 5: Sampling at 350 $\mu$s

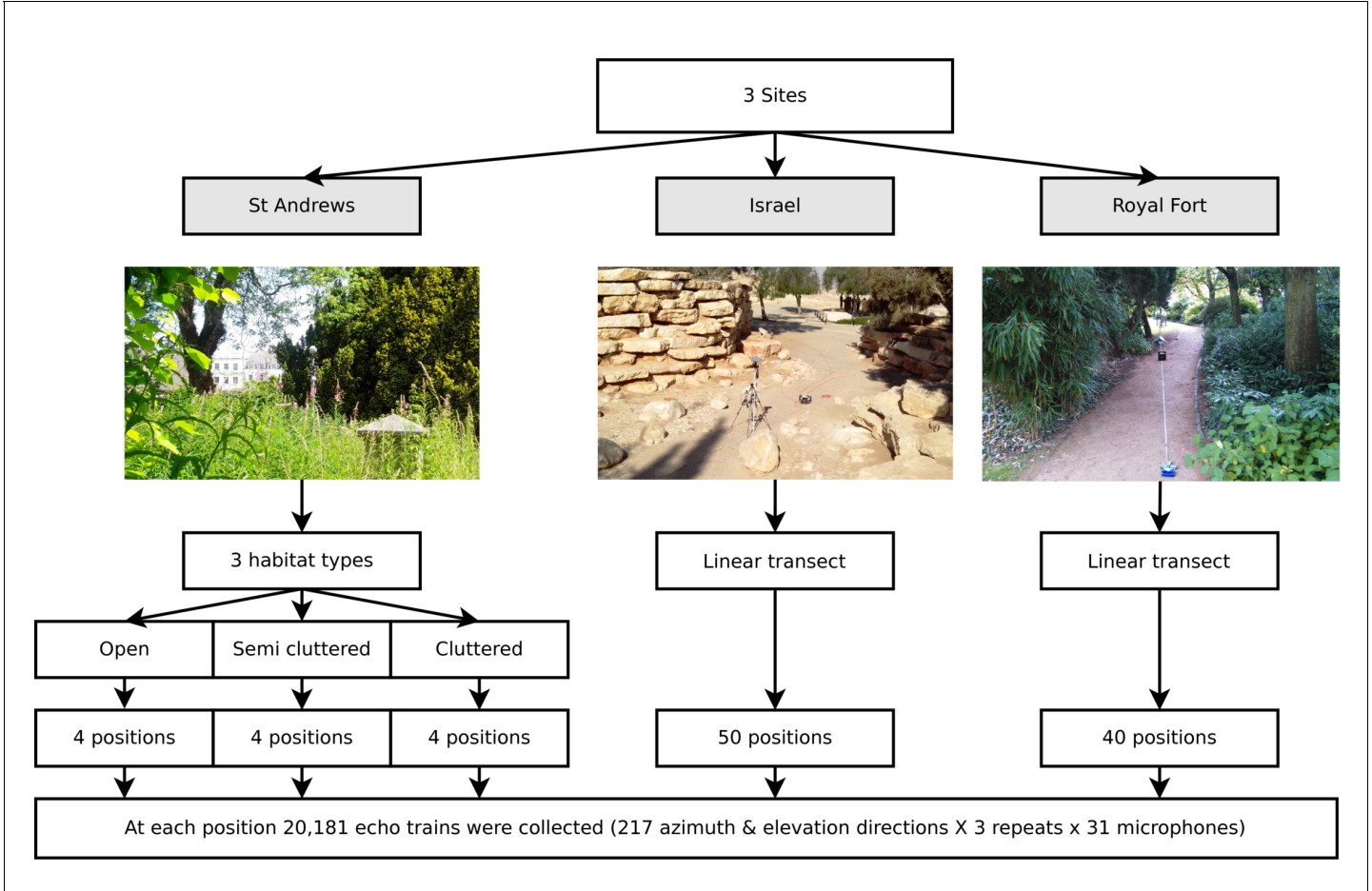

**Figure 3.** Illustration of the data collected. Three data sets were collected (corresponding with three field sites). First, at St Andrews park, the ensonification device was placed at twelve different locations in habitats with varying levels of clutter (Open, Semi Cluttered, and Cluttered). At each of the twelve positions, echo trains from 217 azimuth and elevation directions were collected. At the Israel and Royal Fort Gardens site, the ensonification device was placed at 50 and 40 positions along a straight line, respectively. At each of the 50 (spaced 20 cm apart) or 40 (spaced 25 cm apart) positions, echo trains from 217 azimuth and elevation directions were collected.

8. Step 6: Average across 3 repeats
9. **Output: 217 Templates**

All 2,058,462 echo trains (217 directions × 3 repeats × 31 microphones × 102 positions) were individually filtered using a model of the bats' auditory periphery similar to the one proposed by *Wiegrebe (2008)* to simulate the peripheral hearing system of the bat *Phyllostomus discolor*. In brief, the model consists of a Gammatone filterbank with central frequencies ranging from 30 kHz to 100 kHz in steps of 5 kHz. Subsequently, each channel is exponentially compressed (using an exponent value of 0.4) and low-pass filtered (1 kHz cut-off, 12 dB slope per octave). The cochleogram returned by the model was dechirped by shifting each frequency channel in time (zero-padding at the end) such that the maximum activation, corresponding to the pick up of the emitted signal, is aligned across frequency channels. The first 5.8 ms (corresponding to about 1 m) of the data in each frequency channel were set to zero to avoid including the pick up of the emitted signal or any lingering decay thereof in the analysis. In the model formulated by *Wiegrebe (2008)* a similar operation is performed by means of a channel-wise normalized autocorrelation between call and echoes. However, *Wiegrebe (2008)* worked with simulated calls and echoes. In contrast, in our real measurements, the emission of the pulse saturated the microphones. Hence, we did not have the picked up call available to perform the autocorrelation with. Therefore, we approximated the autocorrelation

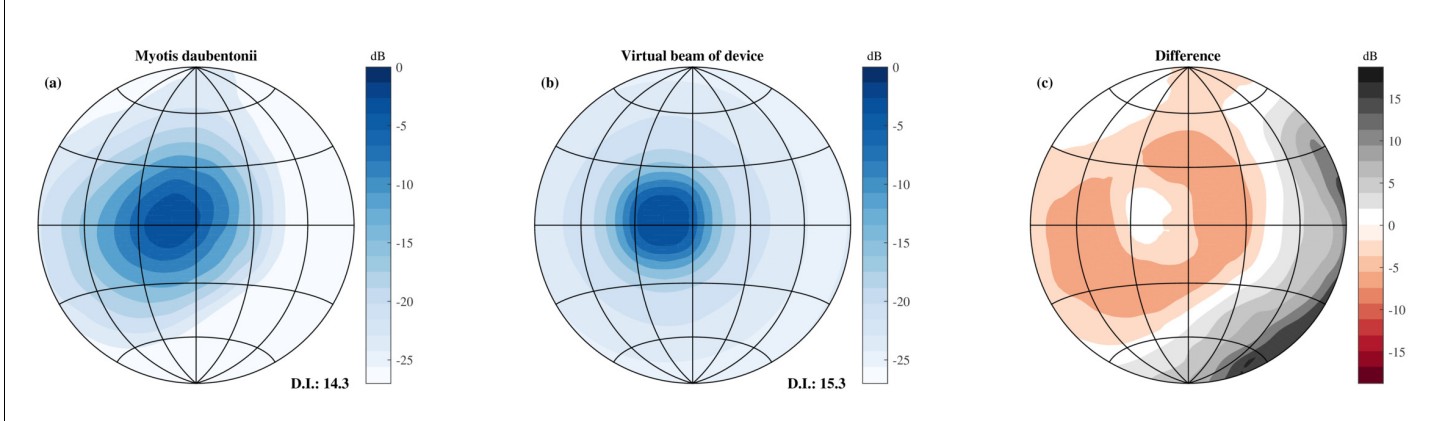

**Figure 4.** Comparison of the simulated beamwidth and the virtual beamwidth of the ensonification device. Comparison of the simulated beamwidth of *M. daubentonii* (panel a) (*Surlykke et al., 2009*) across the range 40 to 100 kHz and the virtual beamwidth of the ensonification device after averaging across 3 neighbouring directions based on the theoretical emission pattern of the Senscomp emitter (panel b). Panel c depicts the difference between the beamwidth of *M. daubentonii* and the virtual beam. Note that for the beam of *M. daubentonii* the emission and hearing directionality were combined.

step in Wiegrebe's model (*Wiegrebe, 2008*) by the dechirping operation described above. Both approaches would result in similar templates.

The resulting cochleograms were averaged across frequencies and the 31 microphones. Hence, the templates used in this paper did not include spectral information. Omitting spectral information reduced the computational complexity of subsequent computations. More importantly, omitting spectral information simplifies the template mechanism (1) making it robust against variations in call design and (2) shows the same mechanism could be used by CF/FM bats as well (see Discussion).

The beam of the emitter used was more narrow than the typically combined hearing and emission directionality of bats, e.g. (*Vanderelst et al., 2010*; *Jakobsen et al., 2013*). Therefore, templates were averaged across 3 neighbouring directions of the pan tilt system in elevation and azimuth to approximate the broader directional sensitivity of bats. Using the theoretical beam directionality of the Senscomp ultrasonic emitter, we estimated the resulting beamwidth employed in this paper to have a 3 dB opening angle of 20 degrees at 55 kHz. *Figure 4* compares the resulting virtual beamwidth of the ensonification device with the simulated beamwidth of the bat *Myotis daubentonii* (*Jakobsen et al., 2013*; *Vanderelst and De Mey, 2008*). From this plot, it is clear that the virtual beam employed in this paper is somewhat more homogeneous than that of actual bats. Nevertheless, the directivity indices of *M. daubentonii* and the virtual beam were very similar (about 14 and 15 *dB* respectively, *Figure 4*). In a next step, the templates were downsampled to a sampling rate of 2.8 kHz, corresponding to a temporal integration interval of 350 $\mu$s. The integration time of the model of *Wiegrebe (2008)* is slightly less than 350 $\mu$s.

Finally, templates were averaged across the three repeats resulting in 217 templates, corresponding to the 217 azimuth and elevation directions for a single location. One template was obtained for each of the 217 directions for each of the 12, 50 or 40 locations at the three respective sites (see *Figure 5*) resulting in a total of 22,134 templates for use in the remainder of the paper.

## Template classification

The classification performance was determined for templates obtained at the St Andrews site. Estimating the probability of correct classification requires the introduction of a noise model. We introduced both a noise floor and stochastic noise on the templates.

### Noise floor

Even in the absence of reflectors returning an echo, the internal noise of the ensonification device resulted in non-zero values for the templates. To avoid template values below the noise threshold of the device to contribute to the classification performance of the templates, we assessed the

template values resulting from measurements taken in absence of any reflector, i.e. the device was pointed upwards in open spaces. These measurements were converted to templates in the same way as described above. The maximum template value obtained from these measurements was taken as the noise threshold $n_f$. Any value in the templates below $n_f$ was set to $n_f$ as that value did not contain any information about the environment beyond the fact that no echo signal was present.

## Stochastic noise

We assume each sample of the templates to be subject to independent Gaussian noise with variance $\sigma_n^2$. The value of $\sigma_n^2$ was derived using a procedure similar to that used by *Dau et al. (1996)*. This is, we determine the value $\sigma_n^2$ that allows discriminating two templates corresponding to two single echo signals differing by 2 dB in intensity (at the 75% correctness criterion). In other words, we assume that the just-noticeable echo intensity difference in bats is 2 dB. Based on the evidence we could find, a just-noticeable-difference of 2 dB is a conservative estimate of the intensity discrimination ability of bats (*Simmons and Vernon, 1971*; *Suthers, 1965*). The noise floor and stochastic noise are illustrated in *Figure 5* by plotting them together with a number of selected templates.

## Template classification performance

Using the noise level $\sigma_n^2$, we calculated the probability $P_c(i)$ for each template $i$ to be correctly classified. This can be expressed as,

$$P_c(i) = P(T_i'|T_i, \sigma_n^2) \tag{1}$$

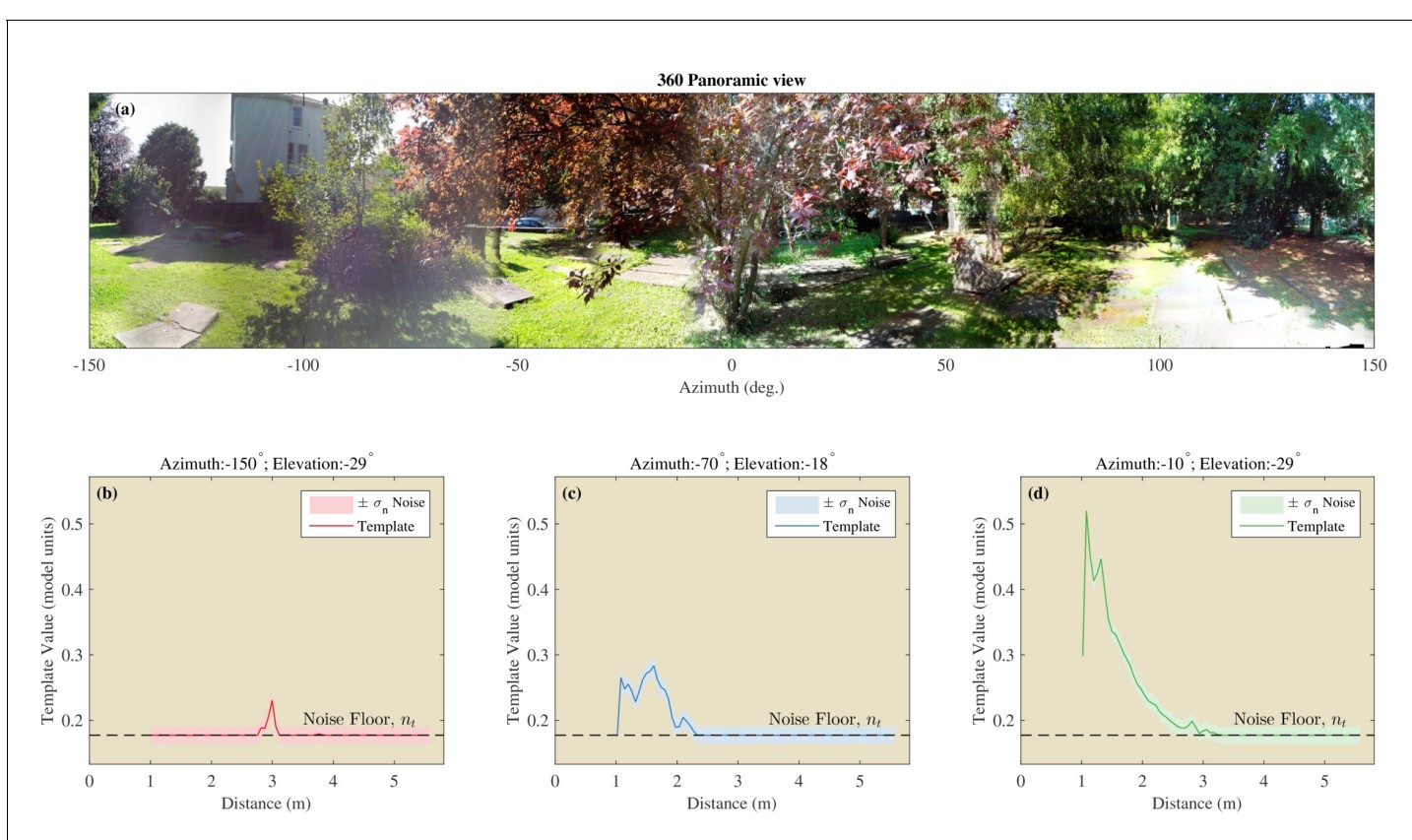

**Figure 5.** Example of the templates using the St Andrews data. (a) 360 degree panoramic view of one of the locations in St Andrews park which was ensonified (Semi cluttered 3). (b–d) Examples of three templates from three different azimuth and elevation directions. The assumed noise $\sigma_n^2$ on the templates is shown by the shaded areas.

with $T_i'$ a measurement originating from the same direction and location as template $i$ but each sample corrupted with noise ($\sigma_n^2$). Therefore, *Equation 1* can be read as the probability of correctly classifying measurement $T_i'$ as originating from template $T_i$ given the noise level $\sigma_n^2$.

$P_c(i)$ is given by the probability that the distance between $T_i$ and $T_i'$ is smaller than the distance between $T_i'$ and any other stored template $T_j$. Formally,

$$P_c(i) = P(d(T_i, T_i') < d(T_j, T_i'))$$ (2)

with $d$ defined as the Mahalanobis distance between template $T_i$ and measurement $T_i'$

$$d(T_i, T_i') = (T_i - T_i')' \times Q^{-1} \times (T_i - T_i')$$ (3)

with $Q$ a diagonal matrix with $\sigma_n^2$ as diagonal elements.

We estimated $P_c(i)$ using a Monte Carlo approach whereby we generated measurements $T_i'$ by adding normally distributed noise to template $T_i$. We calculated the distance between the generated measurement and all templates. $P_c(i)$ was calculated by observing the proportion of generated measurements $T_i'$ that had a smaller distance to $T_i$ than to any other $T_j$. We generated at between 100 and 1000 replications of $T_i'$ for every template. The process of estimating $P_c(i)$ was stopped when the estimate converged and changes to $P_c(i)$ were smaller than 0.01.

In addition to the probability of correct classification we calculated the average angular error $e_i$ as

$$e_i = \sum_c P(T_c | T_i, \sigma_n^2) \cdot g(i, c)$$ (4)

with $g(c, i)$ the great circle distance (i.e. angular separation) between the positions corresponding with templates $T_c$ and $T_i$.

## Quantifying template continuity

### Angular catchment distance

To quantify how smoothly templates change across angles in the St Andrews data set, we calculated the dissimilarity between templates as a function of the angular separation between them. We used the previously defined distance $d(T_i, T_j)$ to calculate the dissimilarity between templates $T_i$ and $T_j$.

For each of the 12 positions $x$, we calculated the average dissimilarity between template $T_x^\phi$ for direction $\phi$ and every other template $T_x^{\phi+\Delta\phi}$ with an angular separation of $\Delta\phi$. The angular catchment distance for each of the 12 positions $x$ was defined as the angular distance $\Delta\phi$ over which the dissimilarity between the templates increased monotonically, in accordance with *Zeil et al., 2003*.

### Linear catchment distance

Template continuity in the Israel and Royal Fort Gardens data sets was quantified in a similar way. We calculated the distance along the corridor over which the dissimilarity (i.e. Mahalanobis distance, *Equation 3*) between templates increased monotonically. For each of the 217 directions $\phi$ and 40 or 50 positions $x$, the dissimilarity between the template $T_x^\phi$ and each template $T_{x+\Delta x}^\phi$ at a different location $x + \Delta x$ (but same direction $\phi$) was calculated using *Equation 3*. Next, the linear catchment distance for every position $x$ was taken as the average distance $\Delta x$ over which the dissimilarity increased monotonically. To allow for the noisy character of the data, we used a 1% threshold. This is, we considered dissimilarity to increase monotonically if any decrease, if present, in dissimilarity was less than 1% of the median dissimilarity found between all pairs of templates in our data set. In addition, we required the dissimilarity to be at least 10% of the median dissimilarity between all pairs of templates in our data. This prevented trivial increases in dissimilarity from being taken into account. Finally, we took the median catchment distance across all positions $x$ for each of the 217 directions $\phi$.

## Results

### Template classification

The probability of correct classification for each of the 217 directions for the 12 positions in the St. Andrews data set is plotted in *Figure 6*. For the Open environments, the probability of correct classification was very low (i.e. $P_c \sim 0$). In contrast, for the Semi cluttered and the Cluttered environments the number of templates that could be recognized with a high probability increased. The fourth Semi cluttered environment was an outlier. The classification probabilities for this position were very low. Physically, this environment might have resembled an Open environment. The branches of the large coniferous tree, the major feature at this position, did not reach low enough for them to generate echoes. *Figure 6* also depicts the angular error $e_i$ for each of the 217 directions in the 12 positions. These results mirror those for the probability of classification showing large errors in the open conditions and small errors in the cluttered positions.

In an additional analysis, we confirmed that only few confusions occurred between templates from different positions. This is, the probability $P_c$ for a given template did not decrease markedly ($\Delta \bar{P}_c \sim 0.01$) when allowing for confusion between a given template (corresponding to one position at one azimuth and elevation direction) and templates taken from other positions (at the same or at a different angle).

### Template continuity

#### Angular catchment area

Using the St. Andrews data, we assessed the angular catchment area for templates by assessing the increase in dissimilarity between templates as a function of angular separation (*Figure 7*). For the templates collected in the Open habitats, we observed virtually no increase in dissimilarity as a function of separation angle. In contrast, for the Semi Cluttered and the Cluttered habitats, there was a tendency for the dissimilarity to increase as a function of angular separation (except for Semi

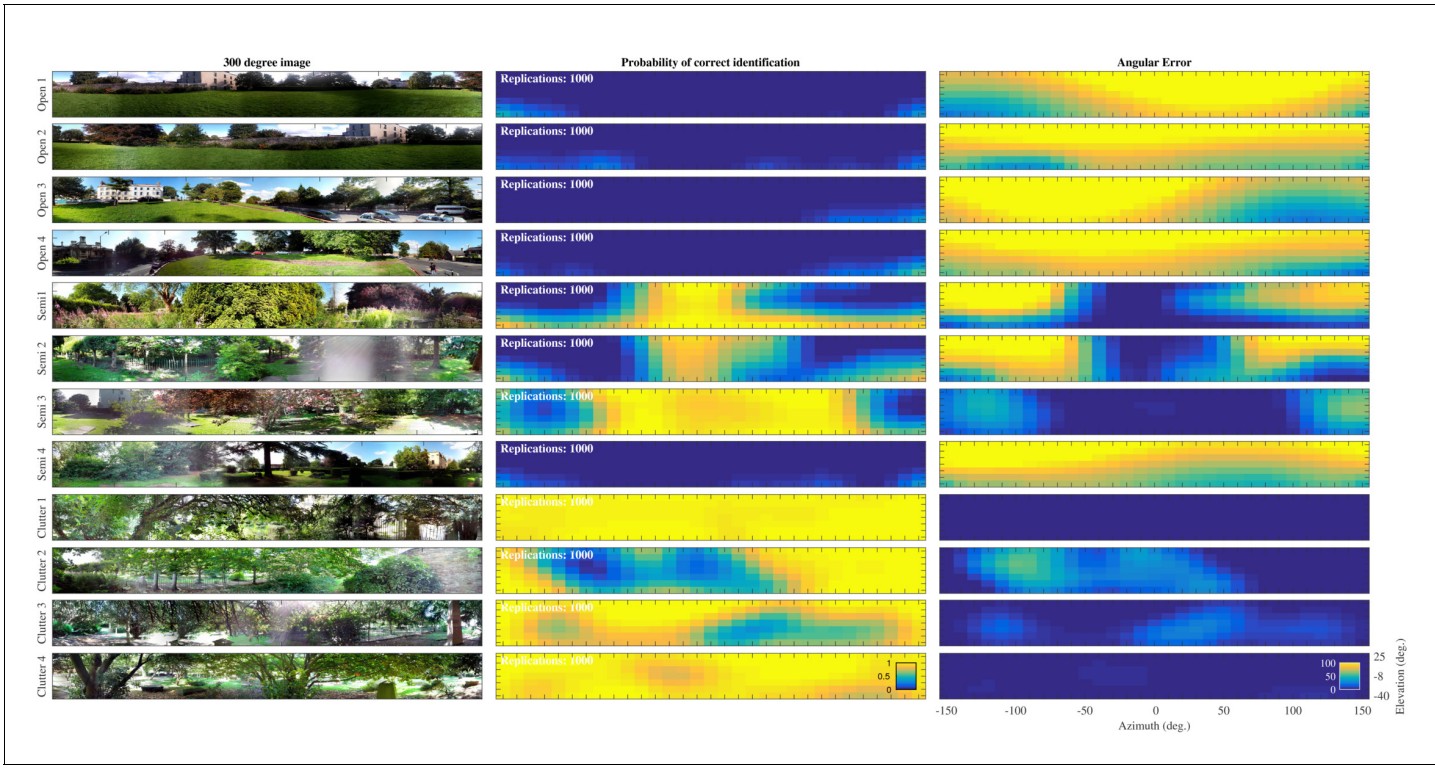

**Figure 6.** Template classification performance. Left: Panoramic views taken at the 12 positions. Middle: Probability ($P_c$) for each of the 12 St Andrews positions as a function of azimuth and elevation. right: Angular error ($e_i$) for each of the 12 St Andrews positions as a function of azimuth and elevation.

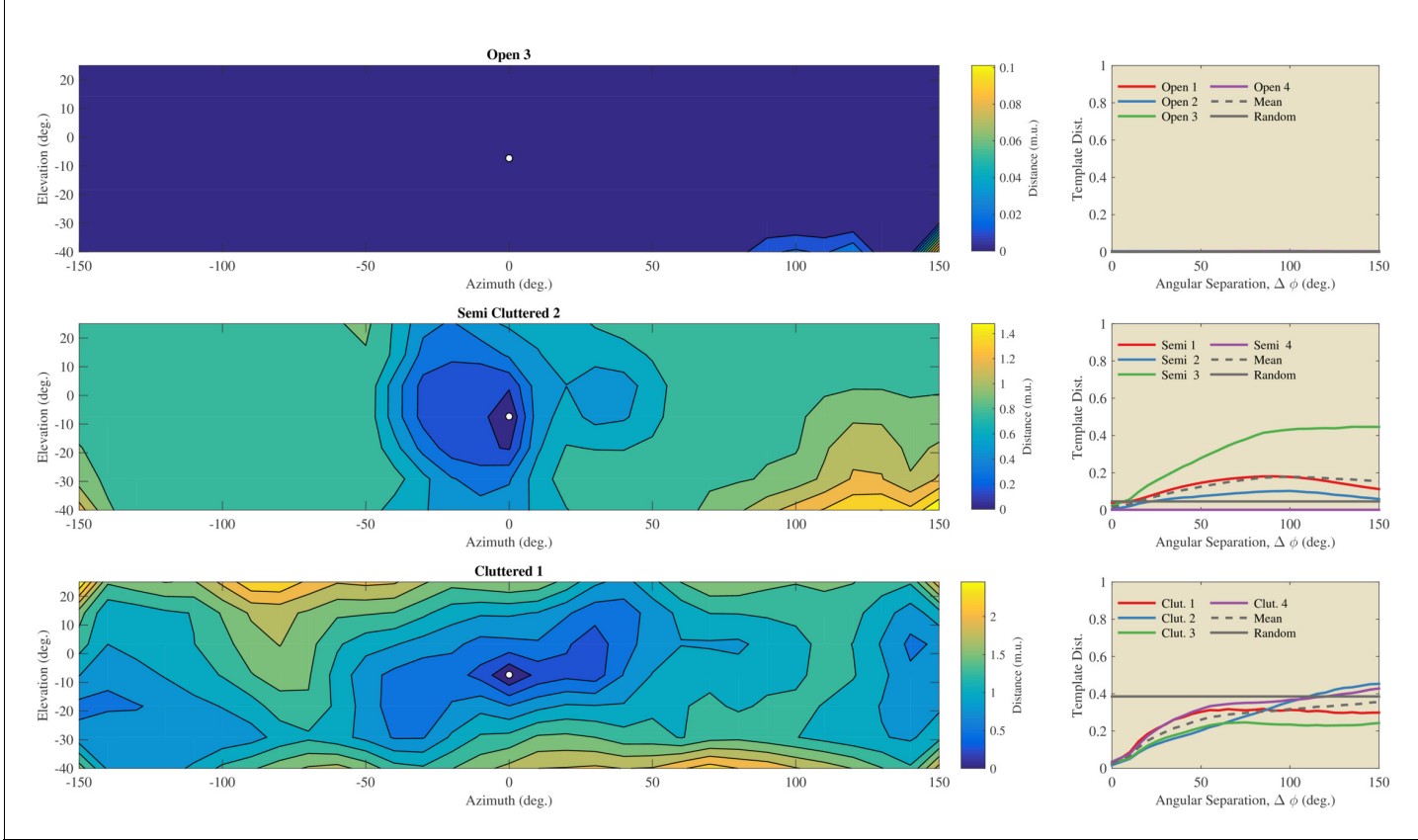

**Figure 7.** Evaluation of the angular catchment areas using the St Andrews data. Left column: Examples of the dissimilarity between a reference template and the other 216 directions for an Open, a Semi Cluttered, and a Cluttered habitat. The direction of the reference template (azimuth 0° & elevation −7°) is indicated by a white dot. The dissimilarity between the reference template and all other templates is depicted by the contour plots. Right: the average dissimilarity between templates as a function of the angular separation in each of the Open, Semi cluttered and Cluttered habitats. The mean dissimilarity as a function of the angular separation for each of the three types of habitats is indicated by a black dotted line. A horizontal black line indicates the average dissimilarity between randomly selected templates. The dissimilarity between unrelated templates serves as a baseline against which the dissimilarity as a function of angular separation can be compared (*Greif and Siemers, 2010*).

Cluttered 4). The angular separation at which the dissimilarity leveled off varied across habitats and habitat types. However, angular catchment areas up to about 90 degrees (Semi 3) and 150 degrees (Cluttered 2) were found. This indicates that, on average, the dissimilarity between templates is a monotonic function of angular separation over a wide range of separation angles.

### Linear catchment distance

*Figures 8a–f* and *9a–f* illustrate the process of assessing the continuity of the Israel and Royal Fort Gardens data sets respectively. In both figures, panels c and f show the linear catchment distances for the templates for two selected azimuth & elevation directions. The plots illustrate the variability in the linear catchment distances across templates with values ranging from approximately 0 to 3.5 m.

In the Israel data, across all positions and directions, the catchment distances rangedup to 6.6 meter. In the Royal Fort Gardens data set, we found catchment distances up to 2.5 meter. Across both data sets the average catchment distance was 0.89 meter (sd: 0.61), see *Figure 10a*. A Wilcoxon rank sum test confirmed that the linear catchment distances for the Israel data were larger than for the Royal Fort Gardens data ($Z = 52.3, p \ll 0.01$). Panels g in *Figures 8* and *9* depict the median linear catchment distance as a function of azimuth and elevation.

We confirmed that for both the Royal Fort Gardens data set and the Israel data set the probability of correct classification was high. The average probabilities $\bar{P}_c$ were 0.82 (sd: 0.21) and 0.99 (sd: 0.03) for the Israel and Royal Fort Gardens data sets, respectively. A Wilcoxon rank sum test

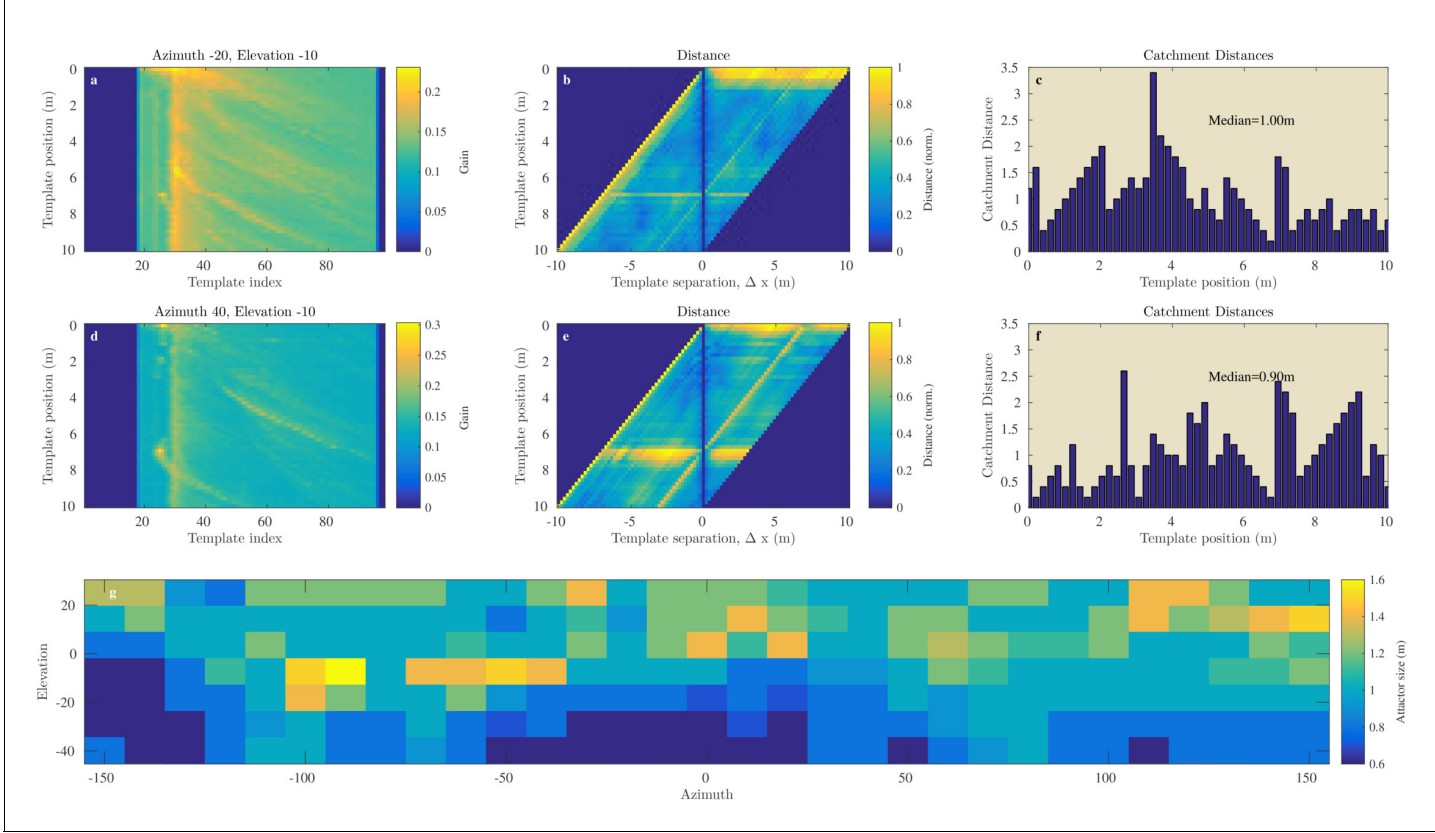

**Figure 8.** Catchment distance for the Israel site. Illustration of the process of finding the catchment distance for the Israel data and the median catchment distance as a function of direction. (**a**) Templates for ensonification direction $-20°°$ azimuth and $-10°$ elevation as a function of the position (i.e. 0 to 10 m). (**b**) The pairwise dissimilarity between the templates in panel (**a**) as a function of the displacement between the templates. (**c**) For each position, a linear catchment distance was calculated. This was done by finding the template separation interval across which the distance increased monotonically. (**c**) The resulting catchment distances for the data in panel (**b**). (**d**), (**e**), (**f**) similar but for azimuth direction 40° and 10° elevation. (**g**) The median catchment distance for each of the 217 azimuth and elevation directions.

confirmed that the templates in the Royal fort gardens were easier to classify correctly. ($Z = -95.2, p \ll 0.01$). The difference in the distribution of $P_c$ for both data sets can also be appreciated from inspecting *Figure 11*.

We calculated the correlation between the linear catchment distance and the probability of correct classification. We found $\rho = -0.18$ (95% C.I.: ±0.02, $p \ll 0.01$) and $\rho = 0.01$ (95% C.I.: ±0.02, $p \gg 0.05$) in the Israel and Royal Fort Gardens data set, respectively. When pooling the data for both sets, a negative correlation between the linear catchment distance and the probability of correct classification was found ($\rho = -0.32$, 95% C.I.: ±0.01, $p \ll 0.01$). The relationship between the probability of correct classification and the linear catchment distance is also depicted in *Figure 11* by means of 2D histograms.

## Discussion

### Template properties

In this paper, we propose template based place recognition might underlie sonar-based navigation in bats. Under this hypothesis, bats would recognize places by remembering their echo signature - rather than their 3D layout. Using ensonification data, we assessed the viability of this alternative hypothesis regarding bat navigation by assessing two critical properties of the templates: (1) unique classification of templates and (2) template continuity. In the following, we discuss our findings regarding these two properties.

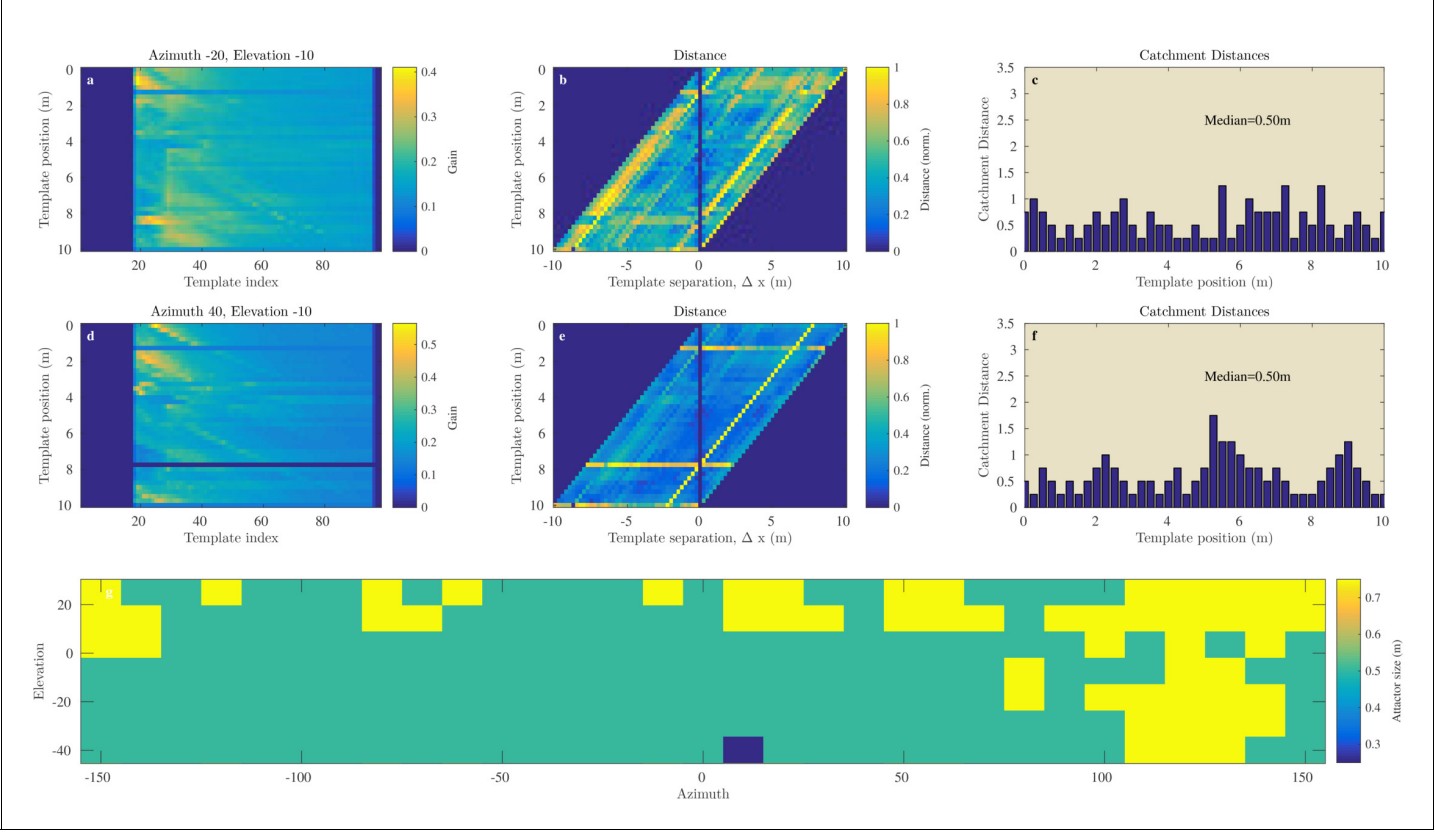

**Figure 9.** Catchment distance for the Royal Fort Gardens data. Similar as **Figure 8** but for the Royal Fort Gardens data set.

## Template classification

The data show that the templates, as constructed in the paper, can be reliably classified, even using a system with a dynamic range smaller than a bat's (**Figure 6**). The typical emission levels of bats are substantially higher (reaching 140 dB$_{spl}$ at 10 cm for a wide frequency range [**Holderied et al., 2005**; **Surlykke and Kalko, 2008**]) than the emission levels for our speaker (maximally 126 dB$_{spl}$ at 50 kHz and 10 cm). Moreover, FM bats have been shown to have hearing thresholds as low as 0 dB$_{spl}$ (**Hoffmann et al., 2008**). In contrast, the Knowles FG series microphones used in the paper have an estimated self-noise level of about 25 dB$_{spl}$ (**Avisoft Bioacoustics, 2015**).

The finding that templates can be readily classified is in agreement with the results of robotic studies that have shown that sonar templates can be used to recognize locations and viewpoints, e.g., (**Steckel and Peremans, 2013**; **Mataric and Brooks, 1990**; **Kuipers, 2000**). By definition, the probability to correctly classify a template depends on the ratio of its systematic variation compared to its random variability. Hence, lacking systematic variation, templates collected in Open habitats could not be classified. As expected, in the absence of diagnostic echoes place recognition is impossible.

It is important to stress that the templates could be correctly classified, even though they did not preserve spectral information. Indeed, the templates were constructed by averaging the cochleograms over the frequencies thereby removing spectral cues (Algorithm 1, step 3). In addition, dechirping of the spectrograms (Algorithm 1, step 1) makes the templates largely independent from call duration. This has two implications. First, navigation by templates as proposed here does not depend heavily on call design. Indeed, the spectral and temporal aspects of bats' calls can vary from call to call, e.g. (**Surlykke and Moss, 2000**; **Kalko, 1995**). By using descriptors of the environment that do not depend critically on call design a bat could navigate the same environment largely independent of the call used. A second implication is that template based navigation should also be feasible for bats using narrowband calls. For example, Rhinolophidae use long narrowband calls preceded and/or followed by a short frequency sweep (**Schnitzler and Denzinger, 2011**). The frequency range of these

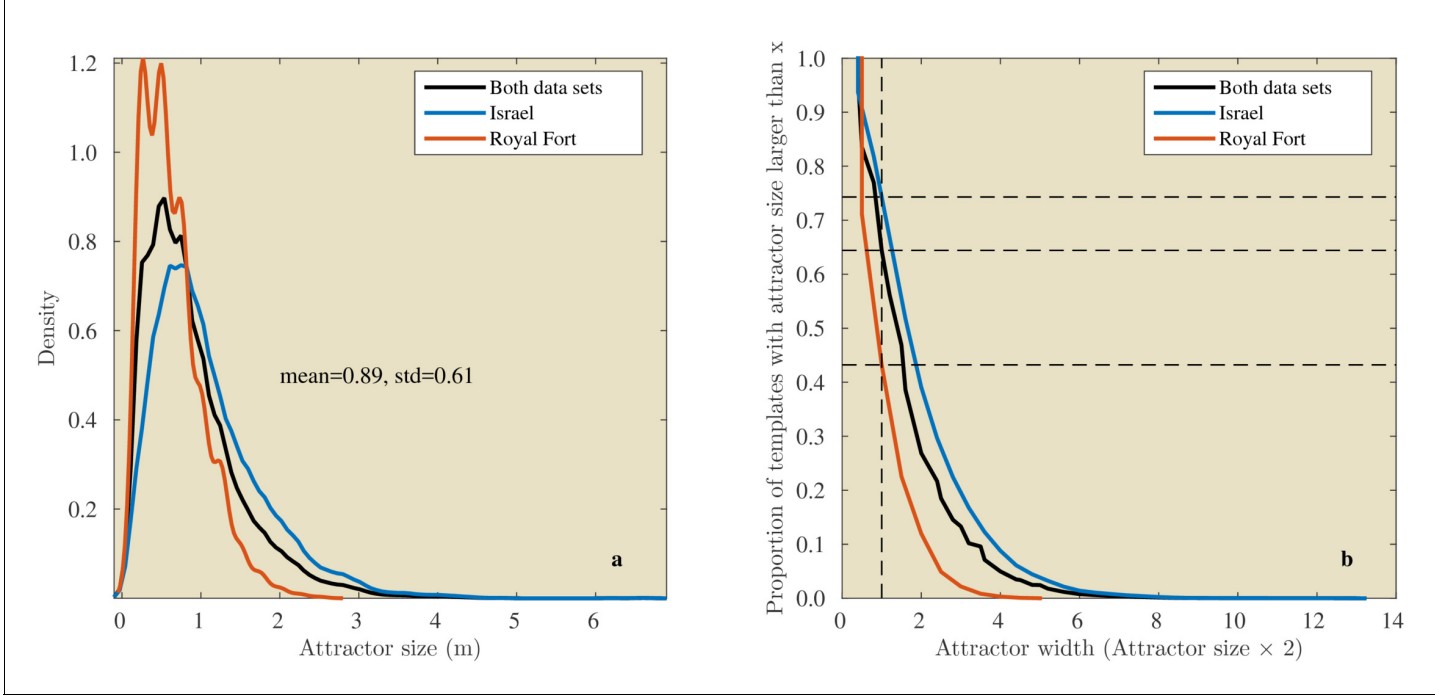

**Figure 10.** Linear attractor distances. (a) Histogram of the linear attractor sizes for both the Israel and Royal Fort Gardens data set. (b) Plot showing the proportion of templates with a linear attractor distance larger than a given value.

sweeps is limited compared to the sweeps used by bats using frequency modulated calls. Nevertheless, Rhinolophidae have specialized in hunting under cluttered circumstances and face rather challenging navigation tasks (*Schnitzler et al., 2003*; *Schnitzler and Kalko, 2001*). A mechanism for recognizing places that does not rely on spectral cues makes for a more plausible candidate for explaining how these bats can find their way using calls with a limited bandwidth.

## Template continuity

We assessed both the angular (St Andrews data, *Figure 7*) and linear (Israel and Royal Fort Gardens data, *Figures 8,9*) catchment areas. We found the dissimilarity between templates to increase monotonically for angular separations up to about 50 and 150 degrees in the Semi Cluttered and Cluttered habits respectively. In addition, on average, the dissimilarity between templates increases monotonically for up about 0.89 m of linear separation (see *Figure 10a*).

Whether these catchment areas are sufficiently large to be functional remains to be tested. However, the linear catchment distances reported here are similar in size to those likely to be experienced by insects using vision based homing. Zeil et al. (*Zeil et al., 2003*; *Stürzl and Zeil, 2007*) collected large sets of panoramic images spaced 10 cm apart in outdoor environments. They calculated the dissimilarity between images as a function of the spacing. They found that linear catchment distances do not exceed 1 m. The catchment distances reported here are also substantially larger than those found in a robotic experiment by *Steckel and Peremans (2013)*. These authors collected sonar based templates in an office environment by driving around a robot equipped with a biomimetic sonar system. They found that displacing the robot by about 14 cm or 20 degrees resulted in leaving the current template's catchment area. In spite of these small catchment areas, they successfully derived a map of the office using the templates. Hence, successful navigation seems to be possible with catchment areas that are smaller than those found in this paper.

In addition, the results regarding the angular and linear catchment distances dovetail with the echolocation behaviour of bats as recorded under field conditions. For many templates, the linear catchment distances are larger than the distance bats seem to cover between calls. *Seibert et al. (2013)* reported a maximum flight speed of about 6 m/s for commuting *Pipistrellus pipistrellus*

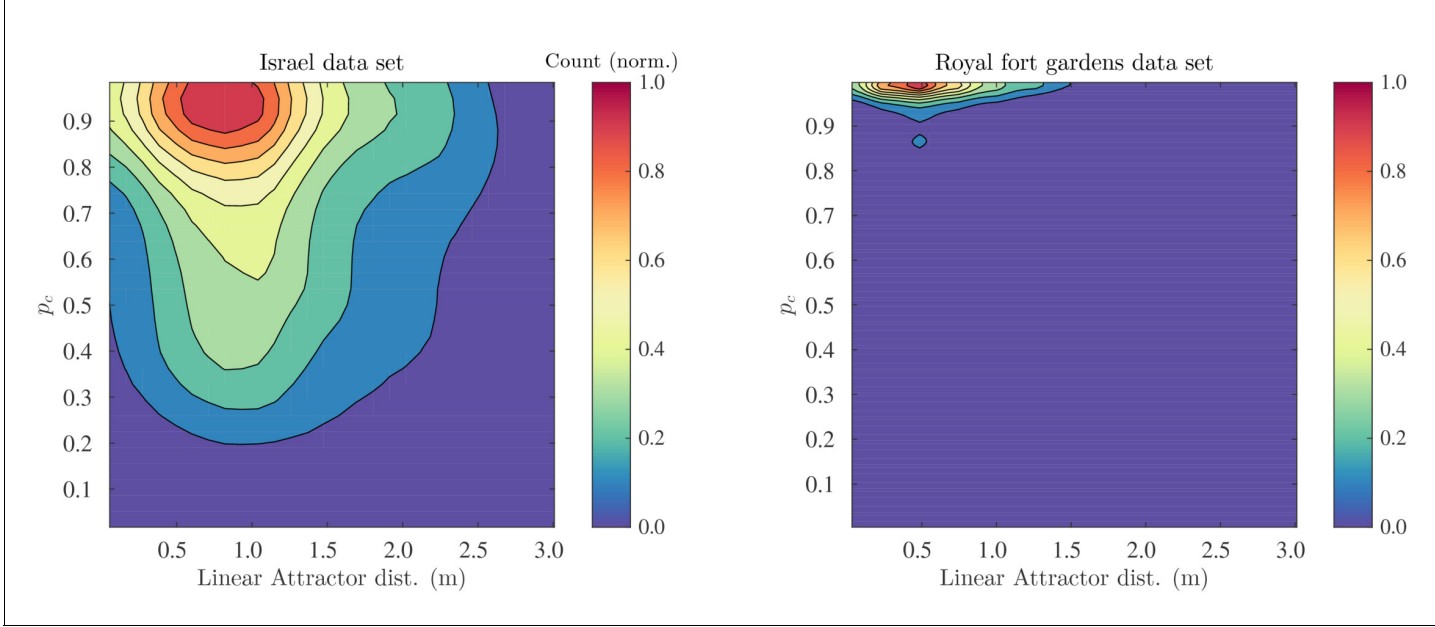

**Figure 11.** Probability of correct classificaiton as a function of the linear catchment distance. 2D histograms of the probability of correct classification $p_c$ versus the linear catchment distances for the templates in the Israel (left) and Royal fort (right) data sets.

across several field conditions. The bats maintained pulse rates higher than 10 Hz. Similar pulse rates and flight speeds were reported for commuting *Myotis dasycneme* (*Verboom et al., 1999*; *Britton et al., 1997*). Finally, commuting *Myotis mystacinus* were found to adopt flight speeds well below 10 ms$^{-1}$ whilst maintaining pulse rates higher than 10 Hz (*Holderied et al., 2006*). In summary, pulse rates higher than 10 Hz seem to be the trend across bat species (*Siemers et al., 2001*) while flight speeds above 10 ms$^{-1}$ are rare (*Holderied, 2001*).

Even a (low) pulse rate of 10 Hz and a (high) flight speed of 10 m/s results in sampling the environment at 1 m intervals. In our data, a substantial portion of templates (about 64%, *Figure 10b*) had a catchment diameter larger than 1 m . Note that the diameter of a catchment area is twice the catchment distance. Therefore, in our data catchment diameters are on average 1.77 m wide (i.e., 2 × 0.89 m). A bat flying through the environment and calling at 10 Hz will always sample the template's catchment region if its diameter is larger than 1 m. Such a template could act as a reliable signpost en route to the bat's destination.

When flying at commuting speeds, bats maintain a low angular velocity. *Holderied (2001)* showed that commuting bats adapt the curvature of their flight path in function of the current flight speed. Even at 5 ms, he found bats to restrict their angular velocity to about 280 deg/s. At a call rate of 10 Hz, this implies a change in direction of about 28 degrees per call. At 10 ms$^{-1}$, this reduces to about 6 degrees per call. Hence, the changes in the orientation of a commuting bat between calls seem to be less than the angular catchment distances found here. Active scanning would result in larger changes in gaze direction. In a study quantifying the scanning behaviour of *Pipistrellus pipistrellus* in the field the largest angle between subsequent call directions was found to be 51 degrees. Hence, this study suggests that, even while actively scanning, bats tend to change the direction of their beam by less than the angular catchment distances of templates in cluttered and semi-cluttered environments. In summary, it seems that the angular spatial sample rate maintained by bats is higher than the average angular catchment distances of templates.

We conclude that the spatial separation of successive calls of bats tends to be less than the size of the catchment areas reported here. In addition, successful navigation is possible with even smaller catchment areas (*Zeil et al., 2003*; *Stürzl and Zeil, 2007*; *Steckel and Peremans, 2013*). Therefore, we propose that the linear and rotational catchment areas reported here are not trivial and are functionally relevant.

The template continuity observed in the current data sets is a consequence of both the beam width and the temporal integration in the model of the auditory periphery. As mentioned in the introduction (*Figure 1*), both parameters determine the volume of space across which echoes are integrated. Wider beams and longer temporal integration result in templates that vary more smoothly with orientation and position in space. For an account of bat navigation that assumes bats to reconstruct a 3D model from an echo train the beam width and temporal integration constitute limitations of the echolocation system by reducing its resolution. In contrast, for a template based approach wider beams and longer integration times are not necessarily a problem. In contrast, these factors might facilitate navigation. As such, the finding that different bat species actively control their emission beams to converge on nearly the same field of view (*Jakobsen et al., 2013*) could indicate that bats perform an optimal spatial smoothing while navigating the environment. Note that, as shown by these authors, optimal fields of view are task and habitat depending resulting in different trade-offs between beam range and width with corresponding linear and angular catchment area sizes.

In summary, we tentatively conclude that the templates satisfy both criteria as put forward in the introduction, i.e. they must allow for unique classification and be sufficiently continuous.

## Differences between sites: possible trade-offs for navigating bats

The Royal Fort Gardens data set resulted in significantly smaller linear catchment distances (*Figures 8*,*9*,*11*). On the other hand, the average probability of correct classification was significantly higher in the Royal Fort Gardens data set (*Figure 1*). The difference in layout between the two sites can be appreciated from inspecting *Figure 3*. The Israel site was a more open habitat than the Royal fort site, where the reflecting foliage was closer to the ensonification device than the rocks at the Royal fort site. In addition, the rocks at the Israel site (in spite of being farther away) resulted in stronger echoes than the foliage at the Royal fort site (compare panels a & d in *Figures 8*,*9*).

The larger linear catchment distances for the Israel data suggests that navigation by means of templates is facilitated by (strong) echoes originating from distant reflectors. Distant reflectors move more slowly through the bat's 'field of view', i.e., they undergo a smaller degree of motion parallax. As our data shows, this results in templates with larger catchment distances. This might facilitate navigation. On the other hand, the Royal Fort Gardens data suggest that the higher motion parallax for closer reflectors results in a better template classification. Indeed, closer reflectors change more abruptly when moving through space. This seems to result in templates that are easier to classify – at the cost of smaller linear catchment distances. In short, our data reveals a trade-off between the linear catchment distance size and the probability of correct classification of a template. The existence of this trade-off is confirmed by the negative correlation we found between the linear catchment distance size and the probability of correct classification.

Hence, our results suggest that navigating bats should maintain a preferred distance to foliage and other reflectors. Indeed, keeping a larger distance would reduce the average object motion parallax and result in larger linear catchment distances. On the other hand, keeping a larger distance would reduce the probability of correctly classifying templates. In addition, flying further away from objects will, in general, result in weaker echoes (in our Israel data this effect has been offset by the strong echoes returned by the rocks). Therefore, our template theory of navigation predicts navigation would be facilitated by keeping a distance from reflectors that balances the opposing effects of motion parallax on the templates' linear catchment distance and probability of correct classification.

## Using templates for mapping

Whilst successful navigation does not require a map-like representation, e.g. (*Cruse and Wehner, 2011*; *Cheung et al., 2014*; *Hesslow, 2012*; *Collett et al., 2013*), it is likely that bats use some sort of cognitive map (*Geva-Sagiv et al., 2015*). However, to the best of our knowledge, the use of such a map has not been demonstrated experimentally (*Holland, 2007*). Nevertheless, indirect evidence for the existence of a cognitive map, integrating echolocation information if available, is the finding of grid and place cells in bats (*Yartsev et al., 2011*; *Yartsev and Ulanovsky, 2013*) and bats' ability to remember flight routes in a completely dark room (*Barchi et al., 2013*).

Building and maintaining a cognitive map based on sonar templates would require the successful completion of three subtasks: (1) exploring the environment and building a library of templates, (2)

integrating the templates into a map-like representation, (3) deriving motor plans from this representation that take the bat from its current location to a desired position. So far, knowledge about how bats could address these challenges is very sparse. However, work has begun on each of these components. Below, we review this work and outline a potential strategy for constructing a cognitive map using the templates as proposed in this paper.

## Exploring

*Vanderelst et al. (2015)* proposed an algorithm for obstacle avoidance in bats that relies on a very simple, yet robust, mechanism comparing the loudness of the onset of the echoes at the left and right ear and turning away from the side receiving the loudest echo. As shown by the simulation results presented by *Vanderelst et al. (2015)*, this simple obstacle avoidance algorithm is able to steer the bat away from obstacles in both planar and true 3D environments. Furthermore, while it contains a stochastic component this obstacle avoidance behaviour still constrains the movement through the environment causing the bat to follow a limited set of routes through a given environment. The results presented here on the sizes of the catchment areas indicate that any sensorimotor strategy that causes the bats to repeatedly visit approximately the same sites will allow them to explore their environment while building a library of templates describing this environment. Hence, this obstacle avoidance mechanism but also other environment driven guidance behaviours (*Geva-Sagiv et al., 2015*), e.g. edge following (*Mataric and Brooks, 1990*; *Verboom et al., 1999*; *Holderied et al., 2006*), that make the bats follow a restricted set of routes through the environment while exploring, could support a template based description of the environment.

## Building the map

Recent work in robotics has offered a suggestion about how bats could integrate templates into a map of the environment. *Steckel and Peremans (2013)* proposed the BatSLAM algorithm to integrate odometry, also called path integration, and templates into a semi-metric map of the environment. BatSLAM is a bio-inspired sonar navigation algorithm, derived from RatSLAM (*Milford et al., 2004*), mirroring the basic functionality of the mammalian hippocampus. In brief, the algorithm directly uses the spectrogram of the echoes as a template to recognize distinct places. The odometry is used to estimate the relative position of different locations labelled by their respective spectrograms. The algorithm averages across measurements of the very noisy odometry for different travels between the same locations. Finally, agreement between conflicting information is sought by applying a relaxation algorithm. The outcome of this process is a graph-like representation of the relative locations of different recognizable places (see *Figure 12*). Details of the mapping algorithm have been presented by *Steckel and Peremans (2013)* and *Wyeth and Milford (2009)*. In summary, BatSLAM, and similar algorithms (*Grisetti et al., 2010*), offer biologically plausible suggestions of how bats can integrate templates and odometry into a semi-metric map of the environment.

Furthermore, we would like to point out that the way such a template based map of the environment is built could also explain the tendency of bats to rely on prominent reflectors or landscape elements as landmarks to guide their navigation (*Verboom et al., 1999*; *Jensen et al., 2005*). This might occur through two different but converging mechanisms. First, prominent reflectors often result in idiosyncratic and thus highly classifiable templates. As BatSLAM uses recognition of previously visited places to drive the relaxation algorithm, the places associated with such templates will act as anchor points for the graph-like map. Indeed, if a place is highly recognizable odometric errors occurring along the various routes that pass through this place will be easier to remove by the relaxation process. Consequently, the metric positions of the routes passing through this place will be defined more precisely relative to the other important places represented in the map, such as start and goal position. A path planner considering this a desirable property would then preferentially generate routes passing through the catchment area associated with this prominent reflector. In our data, highly recognizable templates in the semi-cluttered environments coincided with the locations of shrubs and trees, suggesting their possible use as landmarks. Secondly, prominent reflectors often generate strong echoes making them detectable from a long distance and resulting in the corresponding templates to have large catchment areas (For example, in this paper, the rocks at the Isreal site, *Figures 8,9,10*).

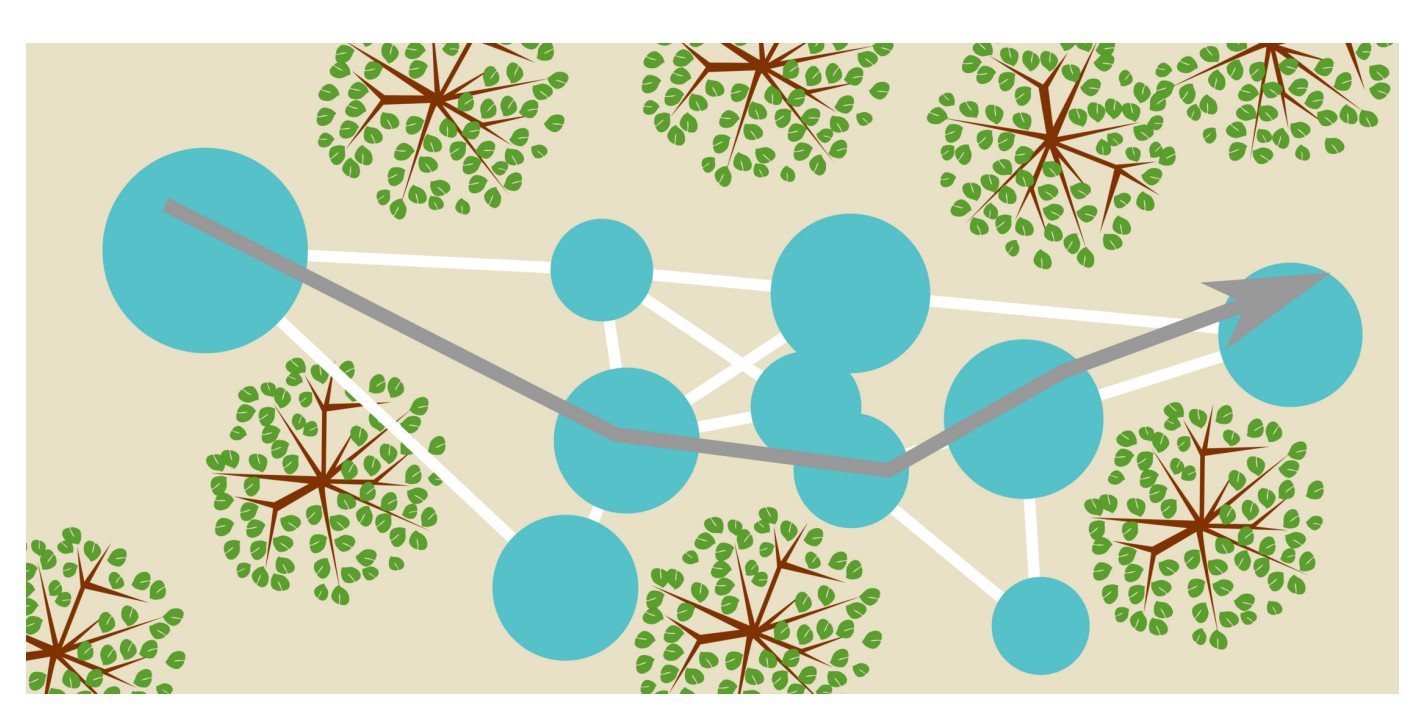

**Figure 12.** Conceptual sketch of an environment and a map. The blue circles represent templates and their catchment areas. The white lines between them represent directions and distances between template locations inferred from odometry, thereby forming a graph-like representation of space. A bat navigating this map (represented by the grey arrow) can use the templates as sign posts at which it reorients itself. Having recognized a particular template (catchment area) the bat knows where it is and approximately which direction it should fly in to arrive at any of the other template locations. Notice that the (catchment areas of) the templates do not cover the complete space.

The extent of the catchment area of a template is positively correlated with the probability that this template will be observed and stored in the map during the exploration of the environment. By biasing the templates included in the map, this mechanism would again bias a planner to generate routes passing through the catchment areas of prominent reflectors. In our data, the linear catchment distances were found to be larger in the Israeli corridor of boulders than in the corridor of vegetation in Royal Fort Gardens, suggesting a possible preference for the use of the former as landmarks. We conclude that a navigation strategy relying on a template based map would show a natural preference for routes including prominent landscape features. Hence, we propose that the use of landmarks might be an emergent feature of a mapping strategy recognizing locations using templates (*Wystrach et al., 2011*).

## Using the map

Using the map requires planning a route and executing it. Of these, planning is the least challenging (*Brooks, 1991*). Many algorithms (*Kuipers, 2000*; *Russell and Norvig, 1995*) have been proposed that allow agents to plan a route given a graph-like network of locations and routes between them. Executing the generated plan requires adequate motor control and is a more difficult problem (*Brooks, 1991*). To the best of our knowledge, no work has been done on how bats select, initiate and execute appropriate sensor-motor loops to use the map to navigate to desired locations (i.e. step 3 in the challenges listed above). In contrast, robotic studies have extensively addressed the issues regarding the execution of a planned sequence of actions (*Kuipers, 2000*; *Siciliano and Khatib, 2008*). We propose that an integrated model of bat navigation could draw inspiration from the solutions derived in this field.

## 3D object layout versus template-based scene representation

In this paper, we have proposed that bats recognize places by remembering their echo signature - rather than their 3D layout. It could be argued that, since bats have been shown to be capable of recognizing objects, they could very well use that same capability to recognize scenes based on the 3D object layout. However, we would like to point out that the behaviour of bats in recognition experiments differs from that of cruising/navigating bats. In recognition experiments, bats typically ensonify the same object from different directions as part of an active object-centred exploration process, e.g. (*Genzel et al., 2012*; *Geipel et al., 2013*; *Simon et al., 2006*). Cruising/navigating bats, on the other hand, fly by objects along their flight path resulting in a more accidental, i.e. less object-centred, and less extensive series of observations of those objects. Such a fly-by mode of echolocation is well suited to the proposed template-based strategy as it requires local 'snapshots' only. Also, we would like to argue that even in the object recognition experiments bats might not be building a 3D reconstruction of the object. They might instead be looking for diagnostic acoustic cues, e.g. spectral cues as hypothesized in the study by *Simon et al. (2006)*. Such an account of object recognition would be easily integrated with our template-based account of place recognition. One possible way the two might interact is that the template-based place descriptors would be used to build a map first and then later (or possibly in parallel) 'landmarks', i.e. uniquely identified objects, would be associated with places on the map.

Hence, while we agree that a 3D place description -were it to exist- would be more general and easier to change, we believe it still remains to be proven that such a description is actually built by bats in a navigation context. Indeed, it is our hypothesis that bats would be driven towards an alternative, template based, mechanism either by the inability of their sonar systems to reconstruct a 3D layout of the environment or, if a 3D layout of the environment could be inferred from the echoes, by the associated much higher computational burden placed upon their limited cognitive resources.

To produce direct evidence that would allow choosing between the two alternatives, we suggest future experiments might be set up to test the prediction by the template based place recognition mechanism advocated here that bats would not be able to distinguish between scenes if their 3D layout is different while their resulting template is similar. Therefore, behavioural evidence in favour of our approach would consist of bats failing to distinguish complex scenes giving rise to many echoes, i.e. with a similar complexity of their natural habitats, that result in the same template, essentially a low-pass filtered range-intensity profile. To test this, one could generate echoes from a complex virtual scene and its mirror image. These should result in the same template whereas a 3D model account predicts bats should be able to distinguish between the two scenes. Alternatively and possibly more straightforward to test, the 3D model account predicts bats would be able to recognize echoes coming from rotated versions of the same underlying scene. The template approach predicts that bats would be unable to make such generalizations for larger rotations, i.e. once the measured template would fall outside the catchment area of the stored template.

## Acknowledgements

We thank Dr. Noam Josef, Gabriella Scatà and Professor Carmi Corine (Ben-Gurion University of the Negev) for their help with data collection in Israel. We thank Rayssa Motta do Nascimento and Ivan Cardoso (Universidade Federal do Rio de Janeiro) for their help with data collection St Andrews park. We thank Sam Riches (University of Bristol) for his help in collecting data at the Royal Park site. Dieter Vanderelst was funded in part by a Postdoctoral Marie Curie Fellowship and by a postdoctoral fellowship from the Research Foundation Flanders.

## Additional information

### Funding

| Funder | Grant reference number | Author |
|---|---|---|
| European Research Council | Marie Curie IEF PIEF-GA-2012-326939 | Dieter Vanderelst |

| Fonds Wetenschappelijk Onderzoek | Dieter Vanderelst |

The funders had no role in study design, data collection and interpretation, or the decision to submit the work for publication.

### Author contributions

DV, MWH, Conception and design, Acquisition of data, Analysis and interpretation of data, Drafting or revising the article; JS, Conception and design, Acquisition of data, Analysis and interpretation of data; AB, Conception and design, Acquisition of data; HP, Conception and design, Analysis and interpretation of data, Drafting or revising the article

### Author ORCIDs

Dieter Vanderelst, http://orcid.org/0000-0001-8049-5178
Andre Boen, http://orcid.org/0000-0002-3489-5063

# Additional files

### Major datasets

The following dataset was generated:

| Author(s) | Year | Dataset title | Dataset URL | Database, license, and accessibility information |
|---|---|---|---|---|
| Dieter Vanderelst, Jan Steckel, Andre Boen, Ma, Herbert Peremans | 2016 | Data and Computer scripts for 'Place recognition using batlike sonar' | http://dx.doi.org/10.5281/zenodo.58579 | Publicly available from the Zenodo website (accession no: 58579) |

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
