## [Decision Letter]

Thank you for submitting your work entitled "Place recognition using batlike bio-sonar" for consideration by *eLife*. Your article has been reviewed by three peer reviewers, and the evaluation has been overseen by a Reviewing Editor and Timothy Behrens as the Senior Editor.

The reviewers have discussed the reviews with one another and the Reviewing Editor has drafted this decision to help you prepare a revised submission.

Summary:

All the reviewers felt there was considerable merit in the study but at the same time had a diversity of major concerns about the some of the assumptions, analyses, and conclusions. The specific comments are provided in detail below.

Reviewer #1:

This article addresses an interesting hypothesis concerning place recognition in biosonar. The authors suggest that to recognize certain locations echolocating bats might employ place templates based on a specific echo signature rather than the 3 dimensional layout of the echo-acoustic scene. By assessing templates' discriminability and continuity independent of the spectral content of the ensonifying signal the proposed model was able to reliably recognize locations allowing which they claim can be sufficient for a successful navigation and orientation in an echo-acoustic environment. Yet, while this paper is intriguing, it might also be pushing the interpretation a little bit too far and should be considered in the larger scope of knowledge about bats, their navigation abilities and what is known so far regarding the link between their neural circuits and echolocation.

The paper refers to "bat echolocation" but different bats have evolved *very* different echolocation strategies to navigate in their environments. Previous studies, also using artificial machines, have shown the capacity of bat-like sonar pulses to extract useful information about the environment, the leap towards suggesting that bats maintain a remarkably large library of pulse signatures to navigate in 3D rather than a memory of a 3D layout (perhaps in the hippocampal formation) is not well supported. From an etiological perspective it is unclear what would drive the system to prefer sonar-based place recognition over 3D spatial memory of the environment as that seems much less of an efficient recognition mechanism and one that is much less easily generalizable and amenable to changes in the environment. The authors also argue in the Discussion that "indirect evidence of a cognitive map exist through findings of grid-cells and place-cells in bats" but in fact those finding are contradicting their claim because those studies have shown that place/grid-cells can exist in the complete absence of echolocation, thus arguing echolocation is not, in fact, necessary for the formation of spatial maps in the hippocampal formation. Nonetheless the paper and the proposed hypothesis in intriguing albeit unlikely capturing the true nature of bat 3D navigation.

Specific comments:

In the second paragraph of the subsection “Model based place recognition”: The authors state that during ensonification of an object spectral cues are generated which encode both location of the object but also general object properties. This induces conflicting information concerning position and shape. This is certainly true but what the authors fail to mention is that bats – along with most mammals – have very flexible and most often disproportionally large ears. This allows them to rapidly move their ears independent of their head and therefore still keeps the outgoing ensonification signal directed toward the object of interest. These rapid ear movements introduce dynamic binaural cues relevant for horizontal object localization but more importantly here also relevant for elevation estimation. Efferent feedback signals encode the ears' position, which is integrated in the evaluation of the incoming echoes. This allows for a better localization performance and thereby can reduce possible conflicts of object shape and position.

In the third paragraph of the subsection “Model based place recognition”: I do not quite agree that spectral cues are degraded when echoes from certain points are temporally integrated due to properties of the auditory system. In fact, distinct spectral cues are generated when temporally peaks are integrated, allowing for object recognition through spectral peak or notch detection (shown in many previous studies). Objects are therefore distinguishable through distinct temporal peaks or when not temporally resolvable through their spectral cues and notches. At the neural level, the work by Jim Simmons further supports this statement. Furthermore, integration can occur at later stages of processing beyond the auditory cortex. While a large fraction of neurophysiological studies have focused on the bat A1, there are many subsequent stages of processing that could lead to a coherent perception of the bat's location in 3D, including the hippocampal formation, basal ganglia, frontal cortex, colliculus, etc.

In the second paragraph of the subsection “Ensonification”: Why was the employed pulse only ranged down to 40 kHz? More echo-acoustic information could have been extracted in lower frequency ranges that could still be perceived by echolocating bats (see publications concerning bat audiograms – albeit the authors do not consider the differences of bat echolocation signals across species). Especially the open space locations might be more reliably recognized by employing lower frequencies that have longer travel times with less atmospheric attenuation.

In the last paragraph of the subsection “Template Construction”: It states that the templates were subsampled at a rate of 350 μs, a rate "slightly larger than the integration time of the model of Wiegrebe". Is subsampling an appropriate way to compensate for integration time? As I understand it, within the integration window accumulated information introduces masking because multiple echoes introduced within that time will be "combined" or considered together (integrated). It seems to me that subsampling would be less akin to integration and more akin to a "refractory period" in which only one echo within the window is taken into consideration, after which no echoes are considered until the end of the window. Would averaging the samples within each 350 μs block be a more appropriate approximation of integration time?

In the first paragraph of the subsection “Template properties”: I was missing a discussion on object recognition based on echolocation sequencing. The authors only state that this is not necessary when employing their model based on echo signature rather than 3D layout. But many studies have shown that bats do indeed integrate sequences of echoes and can thereby reliably discriminate and classify objects.

In the second paragraph of the subsection “Template properties”: the proposed model failed to recognize the echo signatures of the Open environment. The authors state that this is due to the missing diagnostic echoes. Many bat species travel in higher open spaces and still are able to reliably orient themselves. What do the authors propose how bats navigate instead in such an echo-acoustic situation?

In the third paragraph of the subsection “Using templates for mapping”: The authors state that an obstacle avoidance mechanism and other guidance behaviors together with the suggested place recognition template model would allow bats following a restricted set of routes to successfully navigate through their environment. But often new obstacles might come in place disrupting the known echo-acoustic scene. Would the model still be able to reliably recognize the template? It would be interesting by how much echo signature recognition would be disrupted by inserting or deleting temporal characteristics of the templates or by phase warping parts of the signal.

Etiology: In the Abstract and Introduction the authors note that long-distance navigation (Tsoar et al., 2011) requires vision while displaced bats can find their way home from within 15 km by sound (Stones and Branick, 1969 and Williams, Williams and Griffin, 1966). However, the studies cited for these two pieces of evidence refer to distinct species of bats that arose from separate lineages. *Rousettus aegyptiacus* (Tsoar et al., 2011) is a fruit- eating megabat belonging to the same clade as flying foxes and other fruit bats that rely exclusively on vision. It is the only member of its clade to have evolved echolocation in the form of tongue clicks, which are not as sophisticated as the laryngeal echolocation used by microbats, the clade that contains all other echolocating bats including those used in citations [Stones and Branick, 1969] and [Williams, Williams and Griffin, 1966] (*Myotis spp.* and *Phyllostomus hastatus*). The diets and environments of the three species are also quite different. Because the animals evolved to address different environmental pressures it is somewhat misleading to omit the species names in the text, as the findings of one study may not generalize to the species used in the others. The paper never states what genus or even family of bat is most closely modeled by the ensonification device, which is important because many bats have different types of echolocation and foraging strategies. Specifically, which species provided the model for the hyperbolic simulated pulse? Are bats that use calls resembling the simulated pulse found in all three areas ensonified in this study? At what height was the ensonification device placed? Was this the approximate height that bats living in the area might be expected to fly?

Additional data files and statistical comments:

The authors offer to make the data available and I think this is wonderful. The analysis overall seems appropriate.

Reviewer #2:

The current paper uses a combination of biophysical measurements and simulations to present a theoretical framework on how bats may navigate by the auditory analysis of self-generated sounds. Specifically, the authors derive echo-acoustic signatures, templates, from the echoes of their ensonifications and test the extent to which these templates carry information in terms of discriminability and smoothness. The authors hypothesise that templates must be discriminable from each other to meaningfully encode a position and/or orientation within a habitat and it must vary monotonously over a certain range of rotation or translation to be considered as 'smooth'.

Overall the scientific approach is very well conceived and executed. The biophysical measurements are clearly motivated and described and the simulation approach is also well justified. The manuscript would benefit, however, from a more precise wording and better justification of some of the simulation assumptions.

Following are specific comments, in order of appearance, not importance.

In the fourth paragraph of the subsection “Model based place recognition”: The definition and implementation of temporal integration is misleading: temporal integration as a peripheral auditory limitation is (i) a feature of the Gammatone filter bank, determined by the duration of the filter impulse responses and (ii) a feature of the low-pass filtering (cutoff frequency not specified in the current paper) applied after compression. The first integration stage is not really a limit of temporal resolution; it only means that shorter events are recoded onto the frequency axis by the filter bank. The second integration is, I guess, already of a similar order as the 350 µs integration interval applied later. The authors should clarify implicit and explicit integration stages in their model.

In point 1 of the third paragraph of the subsection “Template based place recognition”: The authors use the words 'identify', 'discriminate', 'distinguish' and 'recognize' interchangeably. In psychophysics, (e.g. of object perception) these are quite different levels of perception and so the authors should stick to that expression that matches their simulation, namely discrimination. For identification, for example, discrimination and classification are seen as prerequisites; see e.g. the chapters on object perception in toothed whales in the book by Whitlow Au.

In point 2 of the third paragraph of the subsection “Template based place recognition”: rework definition of smoothness. Only later it becomes clear the monotonicity is required.

In the second paragraph of the subsection “Template Construction”: provide values for compression and low-pass filtering.

In the third paragraph of the subsection “Template Construction”: What is the directionality of the emitting system? If you average cochleograms across microphones, this is quite different than averaging the waveforms. The latter would produce strong directionality but what does averaging the cochleograms produce? This is very unphysiological! Shouldn't you have used the 31 mics like a phased array to thereby impose a bat HRTF on the data?

In the second paragraph of the subsection “Template Construction”: the dechirping appears to be functionally similar to channel wise normalized autocorrelation in Wiegrebe (2008). Why do you deviate from that model here?

In the first paragraph of the subsection “Quantifying discriminability”: you cannot equalize acoustic noise of mics with noise in the template because of non-linear processing (half-wave rectification and compression) in between.

Figure 5: considering that you have unknown emission directionality the comparison with real bats appears not meaningful.

Figure 6: panoramic view!

In the third paragraph of the subsection “Template properties”: CF bats temporal resolution in the CF part however would be very bad (about equal to call duration, 50 ms!) so CF part is not usable, right?

Reviewer #3:

In this manuscript, the authors distinguish between two hypotheses of echolocation-based navigation in bats: model-based place recognition, in which echo cues underlie 3D reconstruction of a scene, vs. template-based recognition, in which a scene is represented directly from cochlear input. The manuscript proposes the viability of the template model, based on analyzing templates constructed from a large body of real-world ensonification data. The criteria for template viability were that a place template had to be discriminable from others, and that it was continuous, i.e. more similar to templates of nearby vs. distant locations. Simulated classification results (using Mahalanobis distance as a discrimination metric) were generally consistent with the clutter of the scenes surveyed. In open spaces with few landmarks, performance was low; in more cluttered spaces performance was high. Additionally, discriminable scenes tended to be continuous. Finally, the authors discuss their results in the context of constructing cognitive maps that can guide mesoscale navigation in bats.

I enjoyed reading the paper and think it should be published. The analyses generally seem sound and carefully performed. Very few statistics are used in the paper; in some places a more formal analysis might lend weight to the results and interpretations. If I am reading the methods correctly, 3 replicate measurements were used to estimate noise at each position, which strikes me as a low number from which to estimate a distribution, whereas the Wiegrebe model used 20.

As an experimentalist I would want to know to what extent bats actually behave in the ways predicted by the reported discrimination probabilities and catchment sizes. For example, the Wiegrebe paper that inspired the model did include some comparisons of *P. discolor* behavior against model predictions. Still, this paper sets the stage nicely for such an evaluation and provides a method for predicting navigation behaviors in a given environment. (I am less convinced that it provides a basis for rejecting object-based navigation mechanisms, which seemed to be an implied aim of the manuscript.)

The authors argue that inherent limitations of biosonar (spectral ambiguity, integration time) make it unlikely that 3D representation of a scene is available to a cruising bat, and that such a representation is therefore unavailable to recognize places. The bases for this argument are plausible, but stop short of ruling out explicit object layout reconstruction as a navigational cue. After all, bats do perform object recognition and localization. Integrating across multiple echo calls, for example, might allow an echolocating bat to ameliorate the spatial blur imposed by the integration time described in the paper. In this vein, the authors might also elaborate on object recognition vs. place recognition "modes" of echolocation. If object-recognition mechanisms are not part of template-based navigation strategy, is there a tradeoff between navigation and object perception performance?

---

## [Author Response]

All the reviewers felt there was considerable merit in the study but at the same time had a diversity of major concerns about the some of the assumptions, analyses, and conclusions. The specific comments are provided in detail below.

Reviewer #1:

*[…] Yet, while this paper is intriguing, it might also be pushing the interpretation a little bit too far and should be considered in the larger scope of knowledge about bats, their navigation abilities and what is known so far regarding the link between their neural circuits and echolocation.*

*The paper refers to "bat echolocation" but different bats have evolved very different echolocation strategies to navigate in their environments.*

The reviewer is right in pointing out that different bat species have evolved different echolocation strategies. However, we believe that the mechanism for place recognition we suggest is general enough to be accessible to most (if not all) echolocating bats. Indeed, as we also point out in the paper, the templates are semi-independent from call duration and frequency content due to the dechirping and averaging across frequencies.

*Previous studies, also using artificial machines, have shown the capacity of bat-like sonar pulses to extract useful information about the environment, the leap towards suggesting that bats maintain a remarkably large library of pulse signatures to navigate in 3D rather than a memory of a 3D layout (perhaps in the hippocampal formation) is not well supported. From an etiological perspective it is unclear what would drive the system to prefer sonar-based place recognition over 3D spatial memory of the environment as that seems much less of an efficient recognition mechanism and one that is much less easily generalizable and amenable to changes in the environment.*

We agree that a number of studies have shown that useful information can be extracted using echolocation. However, the critical point is that no algorithm has shown to be able to reconstruct the 3D layout of the environment from a limited number of incoming echoes – once the scene consists of more than a few (artificial) reflectors. This can be understood by noting that building a 3D description of the environment is an instance of an inverse problem. As such, it comes with all the problems associated with solving inverse problems, i.e. solution methods are ill-posed unless additional relevant constraints can be applied. None of the few regularized solution methods that have been described in the literature (see below: Matsuo, Fontaine, Buck) have been applied to sonar measurements from natural bat environments. A recent study by Dias (see below for reference) found it impossible to reconstruct the layout of a scene consisting of three reflectors.

We conclude that it is still very much an open question whether a 3D description of a natural environment can indeed be constructed from the bat's sonar measurements. Hence, while we agree that a 3D description – were it to exist – would be more general and more easy to change, we also believe it still remains to be proven that such a description is actually built by bats in a navigation context.

It is our hypothesis that either the inability of bat sonar systems to reconstruct a 3D layout of the environment would drive the bats towards an alternative mechanism or, if a 3D layout of the environment could be extracted from the echoes, the bats would still be driven towards a template based approach as inferring the 3D layout would more computational effort. Hence, our computationally less expensive approach would be preferred by evolution as it allows the bats to deal with complex environments in real time.

Ikuo Matsuo, Evaluation of the echolocation model for range estimation of multiple closely spaced objects, J. Acoust. Soc. Am., 130(2), August 2011

Bertrand Fontaine and Herbert Peremans, Determining biosonar images using sparse representations, J. Acoust. Soc. Am., 125(5), May 2009

David A. Hague, John R. Buck, and Igal Bilik, A deterministic compressive sensing model for bat biosonar, J. Acoust. Soc. Am. 132 (6), December 2012

Eduardo Tondin Ferreira Dias, Hugo Vieira Neto. A Novel Approach to Environment Mapping Using Sonar Sensors and Inverse Problems. TAROS 2015: 100-111

The authors also argue in the Discussion that "indirect evidence of a cognitive map exist through findings of grid-cells and place-cells in bats" but in fact those finding are contradicting their claim because those studies have shown that place/grid-cells can exist in the complete absence of echolocation, thus arguing echolocation is not, in fact, necessary for the formation of spatial maps in the hippocampal formation. Nonetheless the paper and the proposed hypothesis in intriguing albeit unlikely capturing the true nature of bat 3D navigation.

We certainly do not want to claim that echolocation is necessary for the formation of a spatial map. All bats will also use visual and idiothetic (and possibly even odor/magnetic field) information to build such maps if available. The only claim we make is that such maps could still be built by a bat that was born blind (and did not have access to those additional sensory cues), i.e. echolocation provides sufficient information to build such maps.

*Specific comments:*

*In the second paragraph of the subsection “Model based place recognition”: The authors state that during ensonification of an object spectral cues are generated which encode both location of the object but also general object properties. This induces conflicting information concerning position and shape. This is certainly true but what the authors fail to mention is that bats – along with most mammals - have very flexible and most often disproportionally large ears. This allows them to rapidly move their ears independent of their head and therefore still keeps the outgoing ensonification signal directed toward the object of interest. These rapid ear movements introduce dynamic binaural cues relevant for horizontal object localization but more importantly here also relevant for elevation estimation. Efferent feedback signals encode the ears' position, which is integrated in the evaluation of the incoming echoes. This allows for a better localization performance and thereby can reduce possible conflicts of object shape and position.*

We agree that the hypothesis presented by the reviewer might explain how bats improve their ability to analyse the layout of the scene. However, to the best of our knowledge, no experimental or computational study has shown that this behaviour allows bats to actually do so. Furthermore, it should be noted that when bats are in the air their relative position with respect to objects of interest is (rapidly) changing all the time. However, a strategy, as proposed by the reviewer, where the bat keeps its relative position to the target fixed and systematically moves its ears to disambiguate spectral cues due to shape from those due to relative position seems feasible only when both bat and target are stationary. Finally, we have studied ear motions similar to the ones described by the reviewer, i.e. those found in the Rhinolophidae, and their potential functionality in detail (see references below analyzing the localization information provided by these motions). This analysis shows that multiple closely-spaced reflectors (as opposed to a single prey target) would prevent the ear movements from introducing reliable 3D cues.

Vanderelst, Dieter, et al. "Information generated by the moving pinnae of Rhinolophus rouxi: tuning of the morphology at different harmonics." PloS one6.6 (2011): e20627.

Vanderelst, Dieter, et al. "Dominant glint based prey localization in horseshoe bats: a possible strategy for noise rejection." PLoS Comput Biol7.12 (2011): e1002268.

*In the third paragraph of the subsection “Model based place recognition”: I do not quite agree that spectral cues are degraded when echoes from certain points are temporally integrated due to properties of the auditory system. In fact, distinct spectral cues are generated when temporally peaks are integrated, allowing for object recognition through spectral peak or notch detection (shown in many previous studies). Objects are therefore distinguishable through distinct temporal peaks or when not temporally resolvable through their spectral cues and notches. At the neural level, the work by Jim Simmons further supports this statement. Furthermore, integration can occur at later stages of processing beyond the auditory cortex. While a large fraction of neurophysiological studies have focused on the bat A1, there are many subsequent stages of processing that could lead to a coherent perception of the bat's location in 3D, including the hippocampal formation, basal ganglia, frontal cortex, colliculus, etc.…*

To avoid confusion about what we mean by in the third paragraph of the subsection “Model based place recognition” we have rephrased the text. We did not mean to suggest that spectral peaks and notches due to non-temporally resolvable echoes from a single reflector are causing interpretation problems, we were suggesting that if such echoes are due to more than one reflector, the resulting spectral peaks and notches complicate the interpretation of both the shape and the location of each individual reflector from that cluster of closely spaced (in terms of distance to bat) reflectors.

Again, as argued in the text, it might be possible to circumvent these interpretation problems by combining consecutive measurements from different points of view (or different pinna orientations) but only at the cost of slowing down environment perception significantly. Hence, we agree with the reviewer that while the mechanism we propose is intended to explain low-level place- recognition and therefore should be fast and effortless (in cognitive terms) this does not preclude the existence of other, slower integration mechanisms at higher cognitive levels to operate in parallel.

In the second paragraph of the subsection “Ensonification”: Why was the employed pulse only ranged down to 40 kHz? More echo-acoustic information could have been extracted in lower frequency ranges that could still be perceived by echolocating bats (see publications concerning bat audiograms – albeit the authors do not consider the differences of bat echolocation signals across species). Especially the open space locations might be more reliably recognized by employing lower frequencies that have longer travel times with less atmospheric attenuation.

Limiting the frequency range to 40 kHz was necessary due to the frequency response properties of the speaker used. A statement to this effect was added to the paper. It is correct that extending the frequency range to lower frequencies would have increased long-range sensitivity and could have made a difference for the open-space templates. However, this would only change what actual environments are classified with the 'open-space' label. No matter what the maximum detection range of a sonar system is, there will always be environments that do not contain reflecting features within that range. Hence, we do not think this would significantly alter the results.

*In the last paragraph of the subsection “Template Construction”: It states that the templates were subsampled at a rate of 350 μs, a rate "slightly larger than the integration time of the model of Wiegrebe". Is subsampling an appropriate way to compensate for integration time? As I understand it, within the integration window accumulated information introduces masking because multiple echoes introduced within that time will be "combined" or considered together (integrated). It seems to me that subsampling would be less akin to integration and more akin to a "refractory period" in which only one echo within the window is taken into consideration, after which no echoes are considered until the end of the window. Would averaging the samples within each 350 μs block be a more appropriate approximation of integration time?*

It seems we were not clear in explaining this aspect of the method. It should be stressed that we perform this sampling on the envelop of the output of the cochlear filter bank, i.e. after low-pass filtering (2nd order Butterworth filter with cut-off frequency 1 kHz). The sample rate is chosen in accordance with Nyquist's sampling theorem. The reason for sampling at this lower rate is to remove the correlations introduced by the integration time of the auditory system. A similar operation to extract templates for object recognition was followed by R. Kuc (see reference below). We have changed the text to make this clear (see also comment by reviewer 2).

Roman Kuc. Biomimetic sonar differentiates coin head from tail. The Journal of the Acoustical Society of America, 101:3198, 1997.

*In the first paragraph of the subsection “Template properties”: I was missing a discussion on object recognition based on echolocation sequencing. The authors only state that this is not necessary when employing their model based on echo signature rather than 3D layout. But many studies have shown that bats do indeed integrate sequences of echoes and can thereby reliably discriminate and classify objects.*

This comment expresses the same concern as a similar one from reviewer 3 and so we repeat our reply there as well. We agree with both reviewers that bats have the ability to recognize objects. However, the behaviour of bats in recognition experiments differs from that of cruising/navigating bats. In recognition experiments, bats typically ensonify the same object from different directions as part of an active object-centered exploration process. Cruising/navigating bats on the other hand, fly by the objects along their flight path resulting in a more accidental, i.e. less object- centered, and less extensive series of observations of those objects. The proposed template-based strategy is well suited to such a fly-by mode of echolocation. Also, we would like to point out that even in the object recognition experiments bats might not be building a 3D reconstruction of the object. They might instead be looking for diagnostic acoustic cues, e.g. spectral cues as hypothesised in the study by Simon et al. Such an account of object recognition would be easily integrated with our template-based account of place recognition. One possible way the two might interact is that the template-based place descriptors would be used to build a map first and then later (or possibly in parallel) 'landmarks', i.e. uniquely identified objects, would be associated with places in the map. However, at this stage this is not much more than conjecture and we have not proposed specific mechanisms for how to do this. Hence, we prefer to keep this topic for future work.

Simon R, Holderied MW, von Helversen O. Size discrimination of hollow hemispheres by echolocation in a nectar feeding bat. J Exp Biol. 2006;209: 3599-3609. doi:10.1242/jeb.02398

*In the second paragraph of the subsection “Template properties”: the proposed model failed to recognize the echo signatures of the Open environment. The authors state that this is due to the missing diagnostic echoes. Many bat species travel in higher open spaces and still are able to reliably orient themselves. What do the authors propose how bats navigate instead in such an echo-acoustic situation?*

As long as a bat still receives echoes from its surroundings the scheme proposed here would apply. If the higher open spaces are so high or so open that the bats do not receive any echoes from their surroundings it is clear that the proposed scheme cannot make use of echolocation input to build maps of those areas. However, in those circumstances, bats would still receive input from other sensing modalities, i.e. visual/magnetic/odor that might provide it with navigation cues in the absence of echolocation cues. A recent review paper by Geva-sagiv discusses the roles of the various sensory systems in bats in relation to the spatial resolution of the navigation task. As we have not studied those alternative sensory systems we have no evidence about the discrimination power of such templates nor about their smoothness. However, we believe it would be very interesting to collect such evidence to see whether our hypothesis still stands or not.

Geva-sagiv M, Las L, Yovel Y, Ulanovsky N. Spatial cognition in bats and rats: from sensory acquisition to multiscale maps and navigation. Nat Publ Gr. Nature Publishing Group; 2015;16: 94–108. doi:10.1038/nrn3888

*In the third paragraph of the subsection “Using templates for mapping”: the authors state that an obstacle avoidance mechanism and other guidance behaviors together with the suggested place recognition template model would allow bats following a restricted set of routes to successfully navigate through their environment. But often new obstacles might come in place disrupting the known echo-acoustic scene. Would the model still be able to reliably recognize the template? It would be interesting by how much echo signature recognition would be disrupted by inserting or deleting temporal characteristics of the templates or by phase warping parts of the signal.*

For now, we have not dealt with the issue of changes in the environment. However, there are a number of ways this model could deal with such changes. One simple way would be to keep track of the prior probability of being currently at template X. This can be done by exploiting the fact that the bat's map would contain topological information, i.e. which templates are next to each other. So, if it is currently at Y it will expect to be in X next when executing a particular flight routine. Such a mechanism is shown to work quite well in an indoor office environment in the biologically inspired BatSLAM algorithm (see reference below). Hence, the bat could have a mechanism that recognizes template X if it is sufficiently similar to the new sensor data and rejects/replaces template X if this data differs too much from the expected template X (based on the a priori probability of being at template X).

Steckel J, Peremans H (2013) BatSLAM: Simultaneous Localization and Mapping Using Biomimetic Sonar. PLoS ONE 8(1): e54076. doi: 10.1371/journal.pone.0054076

*Etiology: In the Abstract and Introduction the authors note that long-distance navigation (Tsoar et al., 2011) requires vision while displaced bats can find their way home from within 15 km by sound (Stones and Branick, 1969 and Williams, Williams and Griffin, 1966). However, the studies cited for these two pieces of evidence refer to distinct species of bats that arose from separate lineages. Rousettus aegyptiacus (Tsoar et al., 2011) is a fruit- eating megabat belonging to the same clade as flying foxes and other fruit bats that rely exclusively on vision. It is the only member of its clade to have evolved echolocation in the form of tongue clicks, which are not as sophisticated as the laryngeal echolocation used by microbats, the clade that contains all other echolocating bats including those used in citations [Stones and Branick, 1969] and [Williams, Williams and Griffin, 1966] (Myotis spp. and Phyllostomus hastatus). The diets and environments of the three species are also quite different. Because the animals evolved to address different environmental pressures it is somewhat misleading to omit the species names in the text, as the findings of one study may not generalize to the species used in the others.*

We agree with the reviewer that mentioning the species in the text would add to the paper's clarity. We have updated the text as such.

The paper never states what genus or even family of bat is most closely modeled by the ensonification device, which is important because many bats have different types of echolocation and foraging strategies. Specifically, which species provided the model for the hyperbolic simulated pulse? Are bats that use calls resembling the simulated pulse found in all three areas ensonified in this study? At what height was the ensonification device placed? Was this the approximate height that bats living in the area might be expected to fly?

As pointed out above, the mechanism proposed is quite independent from the call structure as spectro-temporal cues are removed from the templates. Hence, we do not intend the results to model any particular species. Indeed, independently of the specific echolocation or foraging strategy, the information required to build the templates would be available to every bat that ensonifies its environment at a regular rate. However, as the place-recognition mechanism is intended to make up only the basic layer of the navigation behavior bats are capable of, we agree that specific echolocation or foraging strategies that vary from one bat species to another can be built on top of this capability.

We have added a line stating the height at which the data was collected. While this is somewhat lower than the height at which bats would normally fly, we think the foliage at the level of the measurements to be representative of the foliage at higher positions in the environment.

Reviewer #2:

*Overall the scientific approach is very well conceived and executed. The biophysical measurements are clearly motivated and described and the simulation approach is also well justified. The manuscript would benefit, however, from a more precise wording and better justification of some of the simulation assumptions.*

*Following are specific comments, in order of appearance, not importance.*

*In the fourth paragraph of the subsection “Model based place recognition”: The definition and implementation of temporal integration is misleading: temporal integration as a peripheral auditory limitation is (i) a feature of the Gammatone filter bank, determined by the duration of the filter impulse responses and (ii) a feature of the low-pass filtering (cutoff frequency not specified in the current paper) applied after compression. The first integration stage is not really a limit of temporal resolution; it only means that shorter events are recoded onto the frequency axis by the filter bank. The second integration is I guess already of a similar order as the 350 µs integration interval applied later. The authors should clarify implicit and explicit integration stages in their model.*

We agree with the reviewer about the sources of temporal integration and we have clarified the text to make clear that in our model, in accordance with the Wiegrebe model referred to, the overall temporal integration is determined predominantly by the low-pass filter of the envelop extraction step (2nd order Butterworth filter with cut-off frequency 1 kHz). As explained in our reply to reviewer 1's comment sampling at an interval of 350 µs is done to remove the correlations introduced by the integration time of the auditory system. In response to this comment and a similar comment by reviewer 1, we have now made the rationale behind the sampling clearer. The text now reads as follows:

“In a next step, the templates were sampled at a rate of 1 sample per 350 µs to remove the correlations introduced by the integration time of the auditory system. The integration time of the model of Wiegrebe is slightly less than 350 µs.”

In point 1 of the third paragraph of the subsection “Template based place recognition”: the authors use the words 'identify', 'discriminate', 'distinguish' and 'recognize' interchangeably. In psychophysics, (e.g. of object perception) these are quite different levels of perception and so the authors should stick to that expression that matches their simulation, namely discrimination. For identification, for example, discrimination and classification are seen as prerequisites, see e.g. the chapters on object perception in toothed whales in the book by Whitlow Au.

We thank the reviewer for pointing this out and have reworked the text to be more consistent:

We use the term 'discriminate' to talk about the template discrimination that underlies the place recognition.

We now use the term 'recognize' when talking about places that are to be recognized. In this case, the term 'discrimination' does not feel appropriate.

We use the term 'identification' in the Methods section to describe the mathematical processes. In this context, the term 'identification' is warranted as it is a technical term describing the mathematical operations.

In point 2 of the third paragraph of the subsection “Template based place recognition”: rework definition of smoothness. Only later it becomes clear the monotonicity is required.

The definition was updated accordingly.

*In the second paragraph of the subsection “Template Construction: provide values for compression and low-pass filtering.*

We have added these values.

In the third paragraph of the subsection “Template Construction”: What is the directionality of the emitting system? If you average cochleagrams across microphones, this is quite different than averaging the waveforms. The latter would produce strong directionality but what does averaging the cochleograms produce? This is very unphysiological! Shouldn't you have used the 31 mics like a phased array to thereby impose a bat HRTF on the data?

Detailed specifications of the emitter can be found on the manufacturer's webpage: http://www.senscomp.com/ultrasonic-sensors/series-7000-sensors.php

We would like to emphasize that we do not intend the averaging of cochleograms to model a specific operation in real bats. Averaging across neighboring directions is only intended to mitigate somewhat the high directionality of the speaker used for collecting the echoes. Averaging the cochleograms results in a system with the directionality as plotted in Figure 5.

It is true that, given an omnidirectional speaker, using the 31 microphones as a phased array would allow to approximate the directionality of any bat's echolocation system (emission + HRTF). However, as the speaker used is considerably more directional than the reception-emission apparatus of a bat such an approach would still result in an overall directivity that is considerably higher than the one for the real bat. Furthermore, as our template extraction process removes all spectral cues, approximating a specific HRTF would add very little to our results. Hence, we have preferred to focus on duplicating a realistic ensonification area by averaging the cochleograms.

In the second paragraph of the subsection “Template Construction”: the dechirping appears to be functionally similar to channel wise normalized autocorrelation in Wiegrebe (2008). Why do you deviate from that model here?

Both approaches would result in similar templates. In the Wiegrebe reference an autocorrelation could be performed as in the simulations the call could be 'recorded' without deformation. In contrast, in our real measurements, the emission of the pulse saturated the microphones. Hence, we did not have the picked up call available. We have added this comment to the paper.

In the first paragraph of the subsection “Quantifying discriminability”: you cannot equalize acoustic noise of mics with noise in the template because of non-linear processing (half-wave rectification and compression) in between.

We did not intend to say that the acoustic noise of the microphones was used directly as a measure of the noise for the templates. Hence, it seems that this section was somewhat misleading. As such we have expanded and clarified the way in which σ_2_ was calculated.

Figure 5: considering that you have unknown emission directionality the comparison with real bats appears not meaningful.

The emission directionality of the ensonification device is known as it can be reliably predicted using the directionality equations for a baffled piston. Hence, Figure 5 takes this into account. We have updated the text to make this clear.

Figure 6: panoramic view!

We have corrected the typo.

In the third paragraph of the subsection “Template properties”: CF bats temporal resolution in the CF part however would be very bad (about equal to call duration, 50 ms!) so CF part is not usable, right?

Yes, we assume that CF-FM bats would use the FM part of their calls. We have added this clarification.

Reviewer #3:

[…] I enjoyed reading the paper and think it should be published. The analyses generally seem sound and carefully performed. Very few statistics are used in the paper; in some places a more formal analysis might lend weight to the results and interpretations. If I am reading the methods correctly, 3 replicate measurements were used to estimate noise at each position, which strikes me as a low number from which to estimate a distribution, whereas the Wiegrebe model used 20.

Apparently, we failed to make clear the way in which the σ_2_ was calculated. Indeed, reviewer 2 also commented on this section of the paper. Hence, we have expanded and clarified the text. In brief: we used 12 x 217 templates as input to calculate the value of σ_2_.

As an experimentalist I would want to know to what extent bats actually behave in the ways predicted by the reported discrimination probabilities and catchment sizes. For example, the Wiegrebe paper that inspired the model did include some comparisons of P. discolor behavior against model predictions. Still, this paper sets the stage nicely for such an evaluation and provides a method for predicting navigation behaviors in a given environment. (I am less convinced that it provides a basis for rejecting object-based navigation mechanisms, which seemed to be an implied aim of the manuscript.)

Our mechanism predicts that bats would not be able to distinguish between scenes if their 3D layout is different while their resulting template is similar. Therefore, behavioural evidence in favour of our approach would consist of bats failing to distinguish complex scenes made out of many echoes (i.e. with similar complexity of their natural habitats) that result in the same template, essentially a low-pass filtered range-intensity profile. To test this, one could generate echoes from a complex virtual scene and its mirror image. These should result in the same template whereas a 3D model account predicts bats should be able to distinguish between the two scenes. Alternatively, and possibly more straightforward to test, the 3D model account predicts bats would be able to recognize echoes coming from rotated versions of the same underlying scene. The template approach predicts that bats would be unable to make such generalizations for larger rotations, i.e. once the measured template would fall outside the catchment area of the stored template.

*The authors argue that inherent limitations of biosonar (spectral ambiguity, integration time) make it unlikely that 3D representation of a scene is available to a cruising bat, and that such a representation is therefore unavailable to recognize places. The bases for this argument are plausible, but stop short of ruling out explicit object layout reconstruction as a navigational cue. After all, bats do perform object recognition and localization. Integrating across multiple echo calls, for example, might allow an echolocating bat to ameliorate the spatial blur imposed by the integration time described in the paper. In this vein, the authors might also elaborate on object recognition vs. place recognition "modes" of echolocation. If object-recognition mechanisms are not part of template-based navigation strategy, is there a tradeoff between navigation and object perception performance?*

This comment expresses the same concern as a similar one from reviewer 1 and so we repeat our reply here. We agree with both reviewers that bats have the ability to recognize objects. However, the behaviour of bats in recognition experiments differs from that of cruising/navigating bats. In recognition experiments, bats typically ensonify the same object from different directions as part of an active object-centered exploration process. Cruising/navigating bats on the other hand, fly by the objects along their flight path resulting in a more accidental, i.e. less object-centered, and less extensive series of observations of those objects. The proposed template-based strategy is well suited to such a fly-by mode of echolocation. Also, we would like to point out that even in the object recognition experiments bats might not be building a 3D reconstruction of the object. They might instead be looking for diagnostic acoustic cues, e.g. spectral cues as hypothesised in the study by Simon et al. Such an account of object recognition would be easily integrated with our template-based account of place recognition. One possible way the two might interact is that the template-based place descriptors would be used to build a map first and then later (or possibly in parallel) 'landmarks', i.e. uniquely identified objects, would be associated with places in the map. However, at this stage this is not much more than conjecture and we have not proposed specific mechanisms for how to do this. Hence, we prefer to keep this topic for future work.

Simon R, Holderied MW, von Helversen O. Size discrimination of hollow hemispheres by echolocation in a nectar feeding bat. J Exp Biol. 2006;209: 3599-3609. doi:10.1242/jeb.02398